

# Systematic strong coupling expansion
# for out-of-equilibrium dynamics in the Lieb-Liniger model

Etienne Granet⋆ and Fabian H. L. Essler

The Rudolf Peierls Centre for Theoretical Physics, Oxford University, Oxford OX1 3PU, UK

⋆ etienne.granet@physics.ox.ac.uk

## Abstract

We consider the time evolution of local observables after an interaction quench in the repulsive Lieb-Liniger model. The system is initialized in the ground state for vanishing interaction and then time-evolved with the Lieb-Liniger Hamiltonian for large, finite interacting strength $c$. We employ the Quench Action approach to express the full time evolution of local observables in terms of sums over energy eigenstates and then derive the leading terms of a $1/c$ expansion for several one and two-point functions as a function of time $t > 0$ after the quantum quench. We observe delicate cancellations of contributions to the spectral sums that depend on the details of the choice of representative state in the Quench Action approach and our final results are independent of this choice. Our results provide a highly non-trivial confirmation of the typicality assumptions underlying the Quench Action approach.

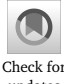

# 1   Introduction

The non-equilibrium dynamics in isolated many-particle quantum systems has attracted a great deal of attention over the last decade [1–6]. These developments were driven by the ability to realize almost isolated many-particle quantum systems using trapped, ultra-cold atoms and investigate their time evolution when driven out of equilibrium in exquisite detail, see e.g. Refs [8–18]. It was realized early on that conservation laws play a crucial role in the

late time relaxational behaviour of isolated systems [9, 19]. This implies in particular that in the thermodynamic limit integrable systems with extensive numbers of conservation laws will typically relax to non-thermal stationary states [20–40]. The full time evolution of local observables in integrable models is equally interesting, but significantly harder to determine. Early work focused on rational conformal field theories [5, 41, 42] and non-interacting models [20, 43–45]. The low density regime after weak quantum quenches has been analyzed by means of linked-cluster expansions [43, 46–49] and semiclassical methods [50–52]. Arguably the method of choice for studying the time evolution of local operators in interacting integrable models is the so-called Quench-Action approach [24, 53]. To date it mostly has been applied to determine and characterize the stationary state [27–30, 33, 35–38]. Exceptions are Refs [47], [54] and [55], which address respectively the asymptotic late-time regimes after quenches to the sine-Gordon, Lieb-Liniger and transverse field Ising models respectively. According to the Quench-Action approach the expectation values of local operators after a quantum quench from an initial state $|\Psi\rangle$ are given by

$$\lim_{L\to\infty} \langle \mathcal{O}(t) \rangle = \lim_{L\to\infty} \left( \frac{\langle \Psi | \mathcal{O}(t) | \Phi_s \rangle}{2\langle \Psi | \Phi_s \rangle} + \frac{\langle \Phi_s | \mathcal{O}(t) | \Psi \rangle}{2\langle \Phi_s | \Psi \rangle} \right). \tag{1}$$

Here $L$ denotes the system size, the so-called representative state $|\Phi_s\rangle$ is a simultaneous eigenstate of the Hamiltonian and of the (quasi)local [34] conservation laws $I^{(n)}$ of the theory under consideration, such that it correctly reproduces the extensive parts of the expectation values of the $I^{(n)}$ in the initial state

$$\lim_{L\to\infty} \frac{\langle \Psi | I^{(n)} | \Psi \rangle}{L} = \lim_{L\to\infty} \frac{\langle \Phi_s | I^{(n)} | \Phi_s \rangle}{L}. \tag{2}$$

The structure of (1) is similar to that of response functions in equilibrium and provides a spectral representation in terms of (normalized) energy eigenstates $|n\rangle$ by writing

$$\langle \Psi | \mathcal{O}(t) | \Phi_s \rangle = \sum_n \langle \Psi | n \rangle \langle n | \mathcal{O}(0) | \Phi_s \rangle \, e^{it(E_n - E_s)}. \tag{3}$$

In practice the Quench Action approach faces two challenges:

- It requires knowledge of the overlaps $\langle \Psi | n \rangle$ between the initial state and energy eigenstates. This is known as the "initial state problem". To date such overlaps are known for a number of specific examples only [56–63], but many of these are physically interesting.

- Determining the time evolution requires carrying out spectral sums like (3). Given that these generally involve an exponentially (in system size) large number of terms this is a formidable challenge.

In this work we focus on the second of these problems, namely how to extract the time dependence of local observables after a quantum quench from the spectral representation. We consider the case of a quantum quench to the repulsive Lieb-Liniger model, and bring to bear strong-coupling expansion methods we recently developed in the context of equilibrium response functions [64].

## 1.1 Lieb-Liniger model

We consider the Lieb-Liniger model [65–67]

$$H = \int_0^L dx \Big[ \psi^\dagger(x) \Big( -\frac{\hbar^2}{2m} \frac{d^2}{dx^2} \Big) \psi(x) + c \psi^\dagger(x) \psi^\dagger(x) \psi(x) \psi(x) \Big], \tag{4}$$

where $\psi(x)$ is a canonical Bose field satisfying equal-time commutation relations

$$[\psi(x), \psi^\dagger(y)] = \delta(x - y).\tag{5}$$

In the following we set $\hbar = 2m = 1$, impose periodic boundary conditions and restrict ourselves to the repulsive case $c > 0$. For later convenience we define the local operators of interest, namely the density operator at position $x$ and the interaction potential

$$\sigma(x) = \psi^\dagger(x)\psi(x)\,,$$
$$\sigma_2(x) = \left(\psi^\dagger(x)\right)^2\left(\psi(x)\right)^2.\tag{6}$$

The Lieb-Liniger model is solvable by the Bethe ansatz [65–67]. Its eigenfunctions can be parametrized by $N$ rapidity variables $\lambda_1, ..., \lambda_N$ that on a ring of radius $L$ satisfy a set of quantization conditions known as "Bethe equations"

$$\frac{\lambda_k}{2\pi} = \frac{I_k}{L} - \frac{1}{L}\sum_{j=1}^{N}\frac{1}{\pi}\arctan\frac{\lambda_k - \lambda_j}{c}\,, \quad k = 1, \dots, N.\tag{7}$$

Here $I_k$ are integer if $N$ is odd and half-odd integer if $N$ is even. The corresponding eigenstate $|\boldsymbol{\lambda}\rangle$ can be written as

$$|\boldsymbol{\lambda}\rangle = B(\lambda_1)...B(\lambda_N)|0\rangle\,,\tag{8}$$

where $B(\lambda)$ is a creation operator acting on a particular reference state $|0\rangle$. The eigenvalues of the Hamiltonian and other conserved quantities are expressed in terms of the rapidities as well. For example the energy $E(\boldsymbol{\lambda})$ and momentum $P(\boldsymbol{\lambda})$ read

$$E(\boldsymbol{\lambda}) = \sum_{i=1}^{N}\lambda_i^2\,, \qquad P(\boldsymbol{\lambda}) = \sum_{i=1}^{N}\lambda_i\,.\tag{9}$$

For $c > 0$ all the solutions $\lambda_i$ to the Bethe equations are real [67].

## 1.2 Quench protocol and observables of interest

Following [29] we consider the following quantum quench protocol. We assume that the system is prepared in the Bose-Einstein condensate (BEC) ground state for $N$ particles in the absence of interactions

$$|\Psi_{\text{BEC}}\rangle = \frac{1}{\sqrt{N!L^N}}\int_0^L dx_1...\int_0^L dx_N\,\psi^\dagger(x_1)...\psi^\dagger(x_N)|0\rangle\,.\tag{10}$$

At $t = 0$ we then suddenly turn on the interactions, so that for $t > 0$ the time evolution of the system $|\Psi(t)\rangle$ is governed by the Hamiltonian (4)

$$|\Psi(t)\rangle = e^{-itH}|\Psi_{\text{BEC}}\rangle\,.\tag{11}$$

Our aim is to determine the full time evolution of a number of different observables after the quench in the framework of the systematic $1/c$-expansion developed in [64]. We have considered the following one and two-point functions:

- One-point function of the interaction potential

$$\langle\sigma_2(0)\rangle_t \equiv \frac{\langle\Psi(t)|\sigma_2(0)|\Psi(t)\rangle}{\langle\Psi(t)|\Psi(t)\rangle}\,.\tag{12}$$

- Density-density correlation function

$$\langle \sigma(x)\sigma(0)\rangle_t \equiv \frac{\langle \Psi(t)|\sigma(x)\sigma(0)|\Psi(t)\rangle}{\langle \Psi(t)|\Psi(t)\rangle}\,. \tag{13}$$

- Steady-state expectation value of the two-point function of the interaction potential

$$\langle \sigma_2(x,\tau)\sigma_2(0,0)\rangle_\infty \equiv \lim_{t\to\infty} \frac{\langle \Psi(t)|\sigma_2(x,\tau)\sigma_2(0,0)|\Psi(t)\rangle}{\langle \Psi(t)|\Psi(t)\rangle}\,. \tag{14}$$

Here we have defined $\sigma_2(x,\tau) = e^{iH\tau}\sigma_2(x)e^{-iH\tau}$. We note that we use a different notation for the time difference $\tau$ to avoid confusion with the time $t$ according to which the system evolves after the quench. The analogous two-point function for the density operator was derived in [64] up to order $1/c^2$.

## 2 Summary of results

As the derivations of our results are quite technical we start by presenting our final answers and discuss their physical implications. All correlators are expressed in terms of distribution functions of particles $\rho(\lambda)$ and holes $\rho_h(\lambda)$ defined as follows [29]

$$\rho(\lambda) = a(\lambda/c)\rho_h(\lambda) = \frac{\tau}{4\pi\big(1 + a(\lambda/c)\big)}\frac{\mathrm{d}a(\lambda/c)}{\mathrm{d}\tau}\,, \tag{15}$$

where $\tau = \frac{\mathcal{D}}{c}$ and

$$a(x) = \frac{2\pi\tau}{x\sinh(2\pi x)}I_{1-2ix}(4\sqrt{\tau})I_{1+2ix}(4\sqrt{\tau})\,, \tag{16}$$

with $I$ the modified Bessel function. The particle density $\mathcal{D}$ is related to $\rho(\lambda)$ by

$$\mathcal{D} = \int_{-\infty}^{\infty}\rho(x)\mathrm{d}x\,. \tag{17}$$

### 2.1 Relaxation of the one-point function $\langle \sigma_2(0)\rangle_t$

Our final result for the time evolution of the interaction potential $\sigma_2(0)$ after the quench, valid at all finite times $t > 0$ and expanded in $1/c$ up to and including order $\mathcal{O}(c^{-4})$ is

$$\langle \sigma_2(0)\rangle_t - \langle \sigma_2(0)\rangle_\infty =$$
$$\lim_{\epsilon\to 0}\frac{16}{c^2(1 + 2\mathcal{D}/c)}\int_0^\infty\int_0^\infty \lambda^2(1 - \tfrac{4\mu^2}{c^2})\cos(2t(\lambda^2 - \mu^2))\rho(\lambda)\rho_h(\mu)e^{-\epsilon\mu^2}\mathrm{d}\lambda\mathrm{d}\mu \tag{18}$$
$$+ \mathcal{O}(c^{-5})\,.$$

The steady-state value $\langle \sigma_2(0)\rangle_\infty$ has been previously calculated in [29]. From (18) the late-time asymptotics can be straightforwardly extracted with a saddle point approximation

$$\langle \sigma_2(0)\rangle_t - \langle \sigma_2(0)\rangle_\infty = \frac{1}{t^3}\frac{\pi}{16(1 + 2\mathcal{D}/c)c^2}\left[\rho(0)\rho_h''(0) - 3\rho''(0)\rho_h(0) - \frac{8\rho(0)\rho_h(0)}{c^2}\right] \tag{19}$$
$$+ \mathcal{O}(t^{-4}) + \mathcal{O}(c^{-5})\,.$$

Here $\rho''(0)$ denotes the second derivative of $\rho(\lambda)$ evaluated at $\lambda = 0$. The asymptotic $t^{-3}$ dependence is in agreement with a previous conjecture [54]. However, our results show that

this regime is reached only at rather late times when the expectation value is already negligibly small. This is shown in Figure 1, where we plot

$$g_2(t) = \frac{\langle \sigma_2(0) \rangle_t}{\mathcal{D}^2}. \tag{20}$$

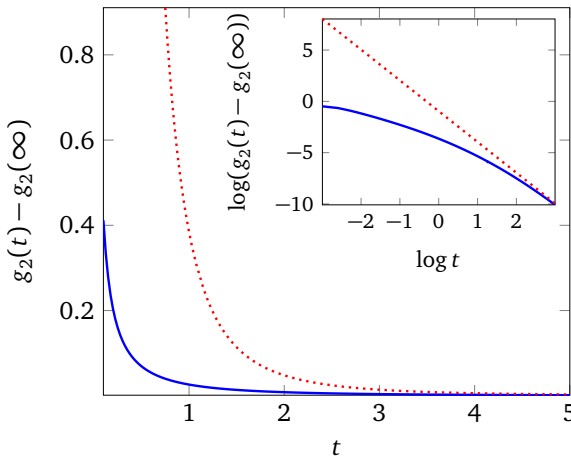

Figure 1: Left: $g_2(t) - g_2(\infty)$ as a function of $t$ (blue thick line), for $c = 3$ and $\mathcal{D} = 0.16$. The dotted red line is the result for the leading asymptotics $\propto t^{-3}$. The inset shows the same quantities on a logarithmic scale.

Our $1/c$-expansion provides us with the first few terms of an expression of the form $g_2(t) = \sum_{n=2}^{\infty} \gamma^{-n} a_n(t)$, where $\gamma = \frac{c}{\mathcal{D}}$ and the functions $a_n(t)$ incorporate non-perturbative summations of certain terms at all orders in $1/c$. In order to assess the parameter range in which the series may be convergent we consider the ratios

$$r_n(t) = \left| \frac{a_n(t)}{a_2(t)} \right|^{\frac{1}{n-2}}, \quad n = 3, 4. \tag{21}$$

In Figure 2 we plots these ratios as functions of $t$ for $c = 3$ and $\mathcal{D} = 0.16$. We see that both

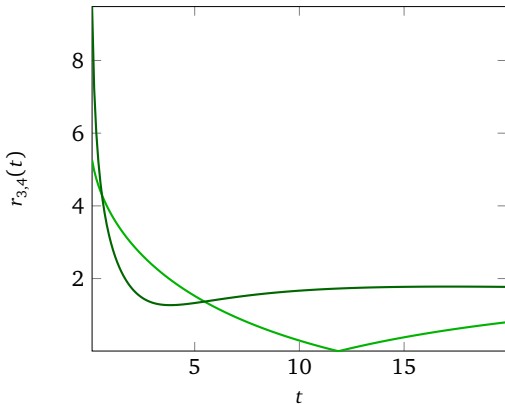

Figure 2: $r_3$ (resp. $r_4$) as a function of $t$, in light (resp. dark) green. These ratios give an estimate of the smallest value of $\gamma$ for which the series is convergent.

ratios grow at short times, indicating that the series is not likely to be uniformly convergent

near $t = 0$. Moreover, it follows from the fact that $g_2(0) = 1$ while $g_2(t) = \mathcal{O}(c^{-2})$ for all $t > 0$ that a resummation of the series is required to capture limit $t \to 0$. For $t \gtrsim 1$ the results for $r_{3,4}(t)$ suggest that the series could be convergent for $\gamma \gtrsim 4$. As a comparison, we recall that the series in $\gamma$ for the ground state energy density is convergent for $\gamma > 4.527$ [68].

## 2.2 Relaxation of the two-point function $\langle \sigma(x)\sigma(0) \rangle_t$

We find that the leading contributions in the $1/c$-expansion of the density-density correlation function can be cast in the form

$$
\begin{aligned}
\langle \sigma(x)\sigma(0) \rangle_t = {} & \langle \sigma(x)\sigma(0) \rangle_\infty \\
& + (1 + 2\mathcal{D}/c)^2 \int_{-\infty}^\infty d\lambda \rho(\lambda) \!\!\!\!\fint d\mu \rho_h(\mu) \frac{\lambda}{\mu} \cos(x'(\mu - \lambda)) \cos(2t(\lambda^2 - \mu^2)) \\
& - \frac{2}{c} \operatorname{sgn}(x) \int_{-\infty}^\infty d\lambda \rho(\lambda) \!\!\!\!\fint d\mu \rho_h(\mu) \frac{\lambda(\lambda - \mu)}{\mu} \sin(x'(\lambda - \mu)) \cos(2t(\lambda^2 - \mu^2)) \\
& + \frac{4}{c} \int_{-\infty}^\infty d\lambda \rho(\lambda) \!\!\!\!\fint d\mu \rho_h(\mu) \!\!\!\!\fint d\nu \rho(\nu) \, F_1(\lambda, \mu, \nu; x') \, \cos(2t(\lambda^2 - \mu^2)) \\
& + \frac{4}{c} \int_{-\infty}^\infty d\lambda \rho(\lambda) \!\!\!\!\fint d\nu \rho(\nu) \!\!\!\!\fint d\mu \rho_h(\mu) \, F_2(\lambda, \mu, \nu; x') \, \cos(2t(\lambda^2 - \mu^2)) + \mathcal{O}(c^{-2}),
\end{aligned}
$$

(22)

where

$$
x' = x \left( 1 + \frac{2\mathcal{D}}{c} \right),
$$

(23)

and

$$
F_2(\lambda, \mu, \nu; x) = \left[ \frac{\lambda(\nu - \lambda)}{\nu(\mu - \nu)} + \frac{\nu - \lambda}{\lambda + \mu} \right] \cos(x(\nu - \lambda)) + \left[ \frac{\nu(\nu - \mu)}{\mu(\lambda - \nu)} + \frac{\nu - \mu}{\lambda + \mu} \right] \cos(x(\nu - \mu)),
$$

$$
F_1(\lambda, \mu, \nu; x) = \left[ \frac{\lambda(\mu - \lambda)}{\mu(\nu - \mu)} + \frac{\lambda(\lambda - \mu)}{\mu(\nu - \lambda)} \right] \cos(x(\mu - \lambda)).
$$

(24)

Here $\fint$ denotes a principal value integral defined as

$$
\fint \frac{f(\lambda)}{\mu - \lambda} d\lambda \equiv \lim_{\epsilon \to 0} \int_{|\lambda - \mu| > \epsilon} \frac{f(\lambda)}{\mu - \lambda} d\lambda.
$$

(25)

The limit $c \to \infty$ of (22) was previously computed in [29]. The density-density correlator (22) is shown in Figs 3 and 4.

We see that for the chosen parameters $\mathcal{D} = 1$ and $c = 10$ the effects of the $\mathcal{O}(c^{-1})$ term are clearly visible and significantly modify the $c = \infty$ result. In particular the oscillatory behaviour as a function of distance for short times becomes more pronounced for smaller values of $c$. Perhaps the most striking feature of Fig. 3 is the apparent absence of any light cone effect [42]. This can be understood by noting that (i) our initial state has an infinite correlation length and any light cone like feature would therefore be weak; (ii) the local Hilbert space is infinite dimensional and the dispersion relation of elementary excitations unbounded. Hence the Lieb-Robinson bound [69] does not apply and "superluminal" effects [70] are allowed.

An alternative representation of (22) more suitable for numerical evaluations and an analysis of the $x \to 0$ and $t \to 0$ limits is presented in Appendix D.

The large $x$ and $t$ asymptotics of (22) at fixed ratio $\alpha = \frac{x}{4t}$ can be determined by a stationary phase approximation, which results in

$$
\langle \sigma(x)\sigma(0) \rangle_t = (1 + \tfrac{2\mathcal{D}}{c})^2 \frac{\pi}{2|t|} \rho(\alpha')\rho_h(\alpha') + o(t^{-1}),
$$

(26)

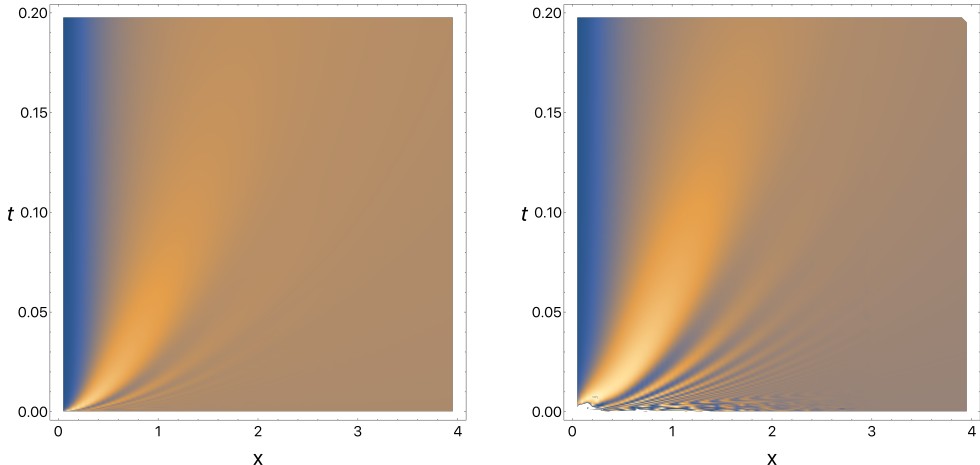

Figure 3: Density plot of $\langle\sigma(x)\sigma(0)\rangle_t$ (22) as a function of $x, t$ for $\mathcal{D} = 1$, $c = \infty$ (left) and $c = 10$ (right). The color coding is the same for both plots.

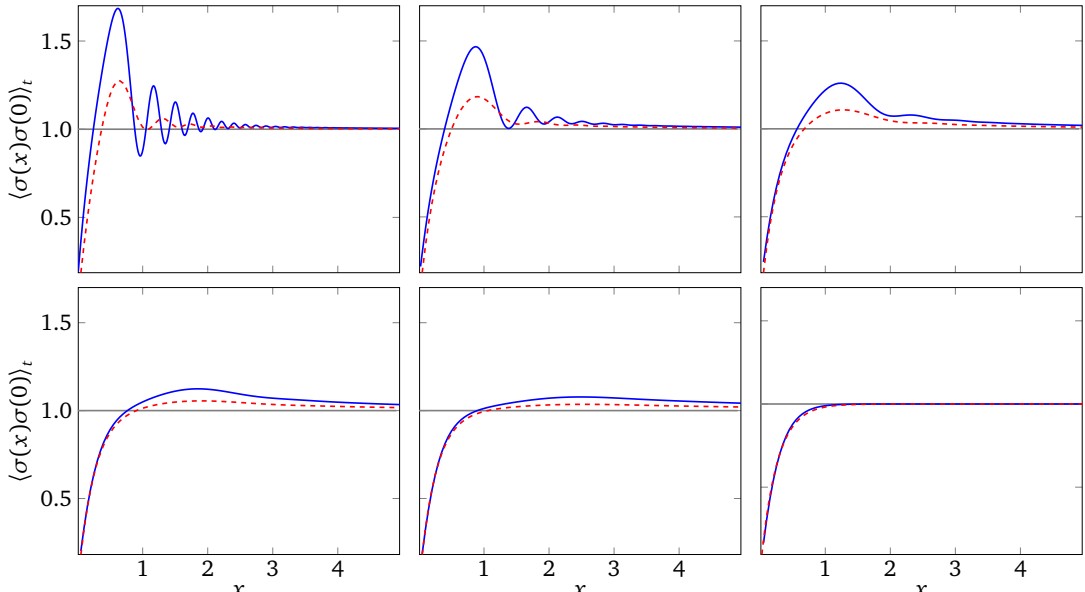

Figure 4: Two-point function $\langle\sigma(x)\sigma(0)\rangle_t$ (22) as a function of $x$, for $\mathcal{D} = 1$, $c = \infty$ (red) and $c = 10$ (blue), for different values of $t = 0.025, 0.05, 0.1, 0.2, 0.3, \infty$ (in reading direction).

with $\alpha' = \frac{x'}{4t}$.

## 2.3 Connected two-point function $\langle\sigma_2(x,\tau)\sigma_2(0,0)\rangle_{\infty,c}$ in the stationary state

We discussed how to determine the non-equal time density-density correlation function in an arbitrary energy eigenstate described by a root density $\rho(\lambda)$ in our previous work [64]. The results in this Section are thus valid for a generic root density $\rho$, the steady state one (15) being a particular case. Applying the same method to the connected dynamical two-point function of $\sigma_2(x)$ gives the following result

$$\langle \sigma_2(x,\tau)\sigma_2(0,0)\rangle_c = \lim_{\epsilon \to 0} \frac{16}{c^4} \int_{-\infty}^{\infty}\int_{-\infty}^{\infty} \rho(\lambda)\rho_h(\mu)\, G(\lambda,\mu)\, e^{i\tau(\lambda^2-\mu^2)+ix(\mu-\lambda)-\epsilon\mu^2}\mathrm{d}\lambda\mathrm{d}\mu$$

$$+ \lim_{\epsilon \to 0} \frac{16}{c^4} \int_{-\infty}^{\infty}\int_{-\infty}^{\infty}\int_{-\infty}^{\infty}\int_{-\infty}^{\infty} \rho(\lambda)\rho_h(\mu)\rho(u)\rho_h(v)(\lambda-u)^2(\mu-v)^2$$

$$\times\, e^{i\tau(\lambda^2-\mu^2)+ix(\mu-\lambda)+i\tau(u^2-v^2)+ix(v-u)-\epsilon\mu^2-\epsilon v^2}\mathrm{d}\lambda\mathrm{d}\mu\mathrm{d}u\mathrm{d}v$$

$$+\, \mathcal{O}(c^{-5}), \tag{27}$$

where we have defined

$$G(\lambda,\mu) = \left[\mathcal{E} + \mathcal{D}\lambda^2 - 2\lambda\mathcal{P} + (\mu-\lambda)(\mathcal{D}\lambda - \mathcal{P})\right]^2,$$

$$\mathcal{P} = \int_{-\infty}^{\infty} \lambda\rho(\lambda)\mathrm{d}\lambda, \qquad \mathcal{E} = \int_{-\infty}^{\infty} \lambda^2\rho(\lambda)\mathrm{d}\lambda. \tag{28}$$

The result for the connected two-point function in the stationary state reached at late times after the quench is obtained by substituting the particle and hole densities (15) into (27) and (28). The leading asymptotic behaviour for large $x$ and $\tau$ with $\alpha = \frac{x}{2\tau}$ kept fixed can be obtained by a stationary phase approximation

$$\langle \sigma_2(x,\tau)\sigma_2(0,0)\rangle_c = \frac{16\pi}{c^4|\tau|}\rho(\alpha)\rho_h(\alpha)(\mathcal{D}\alpha^2 - 2\mathcal{P}\alpha + \mathcal{E})^2 + \mathcal{O}(\tau^{-2}) + \mathcal{O}(c^{-5}). \tag{29}$$

These results can be compared with predictions of Generalized Hydrodynamics (GHD) [71]. According to these the leading large time and distance asymptotics of connected correlations between two local observables $\mathcal{O}_1$ and $\mathcal{O}_2$ is

$$\langle \mathcal{O}_1(x,\tau)\mathcal{O}_2(0,0)\rangle_c = \int_{-\infty}^{\infty} \delta(x - v^{\mathrm{eff}}(\lambda)\tau)\frac{\rho(\lambda)\rho_h(\lambda)}{\rho(\lambda)+\rho_h(\lambda)} V^{\mathcal{O}_1}(\lambda)V^{\mathcal{O}_2}(\lambda)\mathrm{d}\lambda + o(\tau^{-1}), \tag{30}$$

where $V^{\mathcal{O}}(\lambda)$ is the so-called hydrodynamic projection of the operator $\mathcal{O}$, and $v^{\mathrm{eff}}(\lambda)$ the effective velocity associated with the macro-state defined by the particle and hole densities $\rho(\lambda)$ and $\rho_h(\lambda)$. The hydrodynamic projection $V^{\sigma_2}(\lambda)$ of $\sigma_2$ has been determined in [71]

$$V^{\sigma_2}(\lambda) = \frac{2}{\pi}\int_{-\infty}^{\infty} \frac{\rho(\mu)}{\rho(\mu)+\rho_h(\mu)}\frac{g_\mu^{\mathrm{dr}}(\lambda)}{h_1^{\mathrm{dr}}(\lambda)}(h_2^{\mathrm{dr}}(\mu)h_1^{\mathrm{dr}}(\lambda) - h_1^{\mathrm{dr}}(\mu)h_2^{\mathrm{dr}}(\lambda))\mathrm{d}\mu. \tag{31}$$

Here $h_n(\lambda) = \lambda^{n-1}$, $g_\mu(\lambda) = \frac{\mu-\lambda}{(\lambda-\mu)^2+c^2}$, and the dressing operation $h^{\mathrm{dr}}$ is defined by

$$h^{\mathrm{dr}}(\lambda) = h(\lambda) + \frac{1}{2\pi}\int_{-\infty}^{\infty} \frac{2c}{c^2+(\lambda-\mu)^2}\frac{\rho(\mu)}{\rho(\mu)+\rho_h(\mu)}h^{\mathrm{dr}}(\mu)\mathrm{d}\mu. \tag{32}$$

We find that the asymptotics (29) agrees with this GHD prediction at leading order in $1/c$.

The dynamical two-point function of $\sigma_2(x)$ is related to the Drude weight $D$ and the Onsager coefficient $\mathfrak{L}$ by

$$\frac{1}{2}\int_{-\infty}^{\infty} x^2\left[\langle \sigma_2(x,\tau)\sigma_2(0,0)\rangle + \langle \sigma_2(x,-\tau)\sigma_2(0,0)\rangle\right]\mathrm{d}x = D\tau^2 + \mathfrak{L}|\tau| + o(\tau). \tag{33}$$

In contrast to the density-density correlator the two-point function of $\sigma_2(x)$ is expected to exhibit diffusive behaviour, i.e. have a non-vanishing Onsager coefficient $\mathfrak{L} \neq 0$. Our expression for the two point function translates into the following results for $D$ and $\mathfrak{L}$

$$D = \frac{128\pi}{c^4} \int_{-\infty}^{\infty} \lambda^2 \rho(\lambda)\rho_h(\lambda)[\mathcal{E} + \mathcal{D}\lambda^2 - 2\lambda\mathcal{P}]^2 \mathrm{d}\lambda + \mathcal{O}(c^{-5}),$$
$$\mathfrak{L} = \mathcal{O}(c^{-5}). \tag{34}$$

This shows that higher orders in the $1/c$-expansion are required to determine the Onsager coefficient. We note that this specific result holds only for root densities $\rho$ that decay sufficiently fast at infinity. In the case of the steady state root density (15), because of the slow decay of the density, the next terms in the $1/c$ expansion should be re-summed to yield a convergent integral.

# 3 Quench Action approach and $1/c$ expansion

In this Section we discuss the implementation of a $1/c$ expansion of the Quench Action approach [24], which we will then apply to several observables of interest in the remainder of the paper.

## 3.1 The Quench Action approach

The time evolution of the expectation value of any operator $\mathcal{O}$ can always be expressed as a double sum over a basis of energy eigenstates

$$\langle \mathcal{O} \rangle_t = \sum_{\boldsymbol{\lambda}, \boldsymbol{\mu}} \frac{\langle \Psi_{\mathrm{BEC}}|\boldsymbol{\lambda}\rangle \langle \boldsymbol{\lambda}|\mathcal{O}|\boldsymbol{\mu}\rangle \langle \boldsymbol{\mu}|\Psi_{\mathrm{BEC}}\rangle}{\langle \boldsymbol{\lambda}|\boldsymbol{\lambda}\rangle \langle \boldsymbol{\mu}|\boldsymbol{\mu}\rangle} e^{it(E(\boldsymbol{\lambda})-E(\boldsymbol{\mu}))}. \tag{35}$$

Here we have assumed that $\langle \Psi_{\mathrm{BEC}}|\Psi_{\mathrm{BEC}}\rangle = 1$. The Quench Action approach [24] posits that one of the two sums in (35) is completely dominated by states around a saddle point characterized by a certain root distribution $\rho_s$ that is fixed by the overlaps. This allows one to rewrite (35) in the form

$$\lim_{L \to \infty} \langle \mathcal{O} \rangle_t = \lim_{L \to \infty} \frac{1}{|\mathfrak{S}_L|} \sum_{\boldsymbol{\lambda} \in \mathfrak{S}_L} \mathrm{Re}\left[ \sum_{\boldsymbol{\mu}} \frac{\langle \Psi_{\mathrm{BEC}}|\boldsymbol{\mu}\rangle \langle \boldsymbol{\lambda}|\mathcal{O}|\boldsymbol{\mu}\rangle}{\langle \Psi_{\mathrm{BEC}}|\boldsymbol{\lambda}\rangle \langle \boldsymbol{\mu}|\boldsymbol{\mu}\rangle} e^{it(E(\boldsymbol{\lambda})-E(\boldsymbol{\mu}))} \right], \tag{36}$$

i.e. a generalized micro-canonical average [24,72] over a set $\mathfrak{S}_L$ of microstates corresponding to the root density $\rho_s$. Employing typicality ideas the micro-canonical average is then replaced by the expectation value with respect to a single "representative state" $|\boldsymbol{\lambda}\rangle$ [24]

$$\lim_{L \to \infty} \langle \mathcal{O} \rangle_t = \lim_{L \to \infty} \mathrm{Re}\left[ \sum_{\boldsymbol{\mu}} \frac{\langle \Psi_{\mathrm{BEC}}|\boldsymbol{\mu}\rangle \langle \boldsymbol{\lambda}|\mathcal{O}|\boldsymbol{\mu}\rangle}{\langle \Psi_{\mathrm{BEC}}|\boldsymbol{\lambda}\rangle \langle \boldsymbol{\mu}|\boldsymbol{\mu}\rangle} e^{it(E(\boldsymbol{\lambda})-E(\boldsymbol{\mu}))} \right]. \tag{37}$$

We note that this last step assumes that in the thermodynamic limit (37) depends on the representative state $|\boldsymbol{\lambda}\rangle$ only through its root density $\rho(\lambda)$.

## 3.2 "Initial data" for the quench protocol of interest

To be of practical use the representation (37) requires closed-form expressions for the overlaps $\langle \Psi_{\mathrm{BEC}}|\boldsymbol{\lambda}\rangle$. For our quench protocol an efficient representation for the overlaps was derived in [28, 29]. Importantly, the overlaps are non-zero only for "pair" states, i.e. states whose

rapidities are of the form $-\lambda_{N/2}, ..., -\lambda_1, \lambda_1, ..., \lambda_{N/2}$ with $0 < \lambda_j \; \forall j$. We will denote a set of positive $\lambda_j$ by $\boldsymbol{\lambda} > 0$. We will use the notation $\bar{\boldsymbol{\lambda}} = (-\boldsymbol{\lambda}) \cup \boldsymbol{\lambda}$ for such sets of rapidities. The overlaps are then given by

$$\frac{\langle \Psi_{\text{BEC}} | \bar{\boldsymbol{\lambda}} \rangle}{\sqrt{\langle \bar{\boldsymbol{\lambda}} | \bar{\boldsymbol{\lambda}} \rangle}} = (-1)^{N/2} \sqrt{\frac{N!}{L^N}} \sqrt{\frac{\det G^+(\boldsymbol{\lambda})}{\det G^-(\boldsymbol{\lambda})}} \frac{1}{\prod_{j=1}^{N/2} \lambda_j \sqrt{\frac{\lambda_j^2}{c^2} + \frac{1}{4}}}, \tag{38}$$

where $G^{\pm}(\boldsymbol{\lambda})$ are $(N/2) \times (N/2)$ matrices of the form

$$\begin{aligned} G_{ij}^{\pm}(\boldsymbol{\lambda}) = \delta_{ij} & \left( 1 + \frac{1}{L} \sum_{k=1}^{N/2} \frac{2c}{c^2 + (\lambda_i - \lambda_k)^2} + \frac{2c}{c^2 + (\lambda_i + \lambda_k)^2} \right) \\ & - \left( \frac{1}{L} \frac{2c}{c^2 + (\lambda_i - \lambda_j)^2} \pm \frac{1}{L} \frac{2c}{c^2 + (\lambda_i + \lambda_j)^2} \right). \end{aligned} \tag{39}$$

For our quench protocol the saddle point root distribution was determined in Ref. [29] and is given in (15).

## 3.3 The $1/c$ expansion

Our objective is to combine the Quench Action approach to non-equilibrium dynamics (36) with a strong coupling expansion around $c = \infty$. A detailed exposition of the $1/c$ expansion technique for dynamical correlation functions in equilibrium has been given in [64]. In the following we recall the key steps of the method and then extend it to the out-of-equilibrium case.

In order to facilitate the $1/c$-expansion of the form factors and Bethe equations we first fix an arbitrary, large $\Lambda > 0$ that will be sent to $\infty$ at the end of the calculation. We then select an arbitrary averaging state $\boldsymbol{\lambda}$ by fixing its Bethe numbers $\boldsymbol{I}$, impose the constraint that $\forall i, |\lambda_i| < \Lambda$, and define the following overlap-weighted spectral sum

$$\langle \mathcal{O} \rangle_t^{[\boldsymbol{\lambda}], \Lambda} \equiv \text{Re} \left[ \sum_{\substack{\boldsymbol{\mu} \\ \forall i, |\mu_i| < \Lambda}} \frac{\langle \Psi_{\text{BEC}} | \boldsymbol{\mu} \rangle \langle \boldsymbol{\lambda} | \mathcal{O} | \boldsymbol{\mu} \rangle}{\langle \Psi_{\text{BEC}} | \boldsymbol{\lambda} \rangle \langle \boldsymbol{\mu} | \boldsymbol{\mu} \rangle} e^{it(E(\boldsymbol{\lambda}) - E(\boldsymbol{\mu}))} \right]. \tag{40}$$

The overlap-weighted form factor can then be expanded in powers of $1/c$ at fixed $L, \boldsymbol{I}$

$$\frac{\langle \Psi_{\text{BEC}} | \boldsymbol{\mu} \rangle \langle \boldsymbol{\lambda} | \mathcal{O} | \boldsymbol{\mu} \rangle}{\langle \Psi_{\text{BEC}} | \boldsymbol{\lambda} \rangle \langle \boldsymbol{\mu} | \boldsymbol{\mu} \rangle} = \sum_{n=0}^{\infty} \frac{F_n(\boldsymbol{I}, \boldsymbol{J})}{c^n}, \tag{41}$$

where $\boldsymbol{J}$ denotes the Bethe numbers of $\boldsymbol{\mu}$. We also expand the argument of the phase

$$E(\boldsymbol{\lambda}) - E(\boldsymbol{\mu}) = \sum_{n=0}^{\infty} \frac{E_n(\boldsymbol{I}) - E_n(\boldsymbol{J})}{c^n}, \tag{42}$$

but do not expand the phase $e^{it(E(\boldsymbol{\lambda}) - E(\boldsymbol{\mu}))}$ itself in powers of $1/c$. The truncation of the resulting series at a given order $\mathcal{O}(c^{-m})$ defines the $m$-th term of our expansion. Once this truncation has been done, the thermodynamic limit and (if necessary) the average in (36) can be performed. By construction, the result depends only on the root density $\rho$ of the fixed averaging state $\boldsymbol{\lambda}$.

Finally one would like to take the limit $\Lambda \to \infty$. As we will see, the thermodynamic limit of the quantity $\langle \mathcal{O} \rangle_t^{[\boldsymbol{\lambda}], \Lambda}$ at finite $\Lambda > 0$ involves integrals of the form

$$I_n(\Lambda | t, x) = \int_{-\Lambda}^{\Lambda} \mu^n e^{-it\mu^2 + ix\mu} d\mu. \tag{43}$$

The limit $\Lambda \to \infty$ of these integrals for $n > 0$ only exists in a distribution sense, i.e. their integral with any smooth function of $x, t$ has a well-defined limit when $\Lambda \to \infty$. The resulting limits are denoted by $I_n(t, x)$ and have been worked out in [64] for $n = 0, 1, 2$

$$
\begin{aligned}
I_1(t,x) &= \frac{x}{2t} I_0(x,t)\,, \\
I_2(t,x) &= \left( \left( \frac{x}{2t} \right)^2 + \frac{1}{2it} \right) I_0(x,t)\,, \\
I_0(t,x) &= \int_{-\infty}^{\infty} e^{-it\mu^2 + ix\mu} \mathrm{d}\mu\,.
\end{aligned}
\tag{44}
$$

An equivalent representation is

$$
I_n(t,x) = \lim_{\epsilon \to 0} \int_{-\infty}^{\infty} \mu^n e^{-it\mu^2 + ix\mu - \epsilon\mu^2} \mathrm{d}\mu\,.
\tag{45}
$$

The process described above provides closed-form expressions at order $\mathcal{O}(c^{-m})$ for the quantities

$$
\langle \mathcal{O} \rangle_t^{[\rho]} \equiv \lim_{\Lambda \to \infty} \lim_{L \to \infty} \frac{1}{|\mathfrak{S}_L|} \sum_{\boldsymbol{\lambda} \in \mathfrak{S}_L} \langle \mathcal{O} \rangle_t^{[\boldsymbol{\lambda}], \Lambda}\,.
\tag{46}
$$

Finally, in order to obtain the out-of-equilibrium time evolution (36) this result needs to be evaluated for the saddle point root density $\rho$ describing the quench protocol of interest.

## 4 Calculation of the one-point function $\langle \sigma_2(0) \rangle_t$

In this Section we apply the Quench Action approach combined with a $1/c$ expansion to compute the one-point function $\langle \sigma_2(0) \rangle_t$.

### 4.1 The form factors

In order to evaluate the expression (36), one requires a closed-form expression for the form factors of $\sigma_2$ between energy eigenstates. In the case of interest, because of the structure of the non-vanishing overlaps with the initial state $|\Psi_{\mathrm{BEC}}\rangle$, the states entering (36) have a pair structure and will be denoted $|\bar{\boldsymbol{\lambda}}\rangle$ and $|\bar{\boldsymbol{\mu}}\rangle$. Hence they have same (vanishing) momentum. In this situation the normalized form factors have been calculated previously and read [79]

$$
\begin{aligned}
\frac{\langle \boldsymbol{\mu} | \sigma_2(0) | \boldsymbol{\lambda} \rangle}{\sqrt{\langle \boldsymbol{\lambda} | \boldsymbol{\lambda} \rangle \langle \boldsymbol{\mu} | \boldsymbol{\mu} \rangle}} = {} & \frac{(-i)^{N+1}(-1)^{N(N-1)/2}}{2cL^N \sqrt{\det G(\boldsymbol{\lambda}) \det G(\boldsymbol{\mu})}} \frac{(E(\boldsymbol{\mu}) - E(\boldsymbol{\lambda}))^2}{N} \prod_{j \neq p} (V_j^+ - V_j^-) \\
& \times \frac{\prod_{i<j} |\lambda_i - \lambda_j| \prod_{i<j} |\mu_i - \mu_j|}{\prod_{i,j} (\lambda_i - \mu_j)} \sqrt{\prod_{i,j} \frac{\lambda_i - \lambda_j + ic}{\mu_i - \mu_j + ic}} \det_{i,j=1,\dots,N} \left[ \delta_{ij} + U_{ij} \right].
\end{aligned}
\tag{47}
$$

Here $1 \le p \le N$ is an arbitrary integer and

$$V_i^\pm = \prod_{k=1}^N \frac{\mu_k - \lambda_i \pm ic}{\lambda_k - \lambda_i \pm ic},$$

$$U_{jk} = \frac{i}{V_j^+ - V_j^-} \frac{\prod_m (\mu_m - \lambda_j)}{\prod_{m \neq j}(\lambda_m - \lambda_j)} \left( \frac{2c}{c^2 + (\lambda_j - \lambda_k)^2} - \frac{2c}{c^2 + (\lambda_p - \lambda_k)^2} \right)$$

$$+ \frac{i}{V_j^+ - V_j^-} \frac{2c}{c^2 + (\lambda_p - \lambda_k)^2},$$

$$G(\boldsymbol{\lambda})_{ij} = \delta_{ij} \left( 1 + \frac{1}{L} \sum_{k=1}^N \frac{2c}{c^2 + (\lambda_i - \lambda_k)^2} \right) - \frac{1}{L} \frac{2c}{c^2 + (\lambda_i - \lambda_j)^2}. \tag{48}$$

## 4.2  $1/c$ expansion and particle-hole excitations

Employing a saddle-point argument in (37) shows that in the limit $t \to \infty$ we have

$$\langle \mathcal{O} \rangle_\infty \equiv \lim_{t \to \infty} \lim_{L \to \infty} \langle \mathcal{O} \rangle_t = \lim_{L \to \infty} \mathrm{Re} \frac{\langle \bar{\boldsymbol{\lambda}} | \mathcal{O} | \bar{\boldsymbol{\lambda}} \rangle}{\langle \bar{\boldsymbol{\lambda}} | \bar{\boldsymbol{\lambda}} \rangle}. \tag{49}$$

We use this and the pair structure of the states entering (37) to rewrite (37) as

$$\lim_{L \to \infty} \langle \mathcal{O} \rangle_t = \langle \mathcal{O} \rangle_\infty + \lim_{L \to \infty} \mathrm{Re} \sum_{\substack{\boldsymbol{\mu} > 0 \\ \boldsymbol{\mu} \neq \boldsymbol{\lambda}}} \frac{\langle \Psi_{\mathrm{BEC}} | \bar{\boldsymbol{\mu}} \rangle \langle \bar{\boldsymbol{\lambda}} | \mathcal{O} | \bar{\boldsymbol{\mu}} \rangle}{\langle \Psi_{\mathrm{BEC}} | \bar{\boldsymbol{\lambda}} \rangle \langle \bar{\boldsymbol{\mu}} | \bar{\boldsymbol{\mu}} \rangle} e^{2it(E(\boldsymbol{\lambda}) - E(\boldsymbol{\mu}))}. \tag{50}$$

We now analyze this expression in terms of a $1/c$-expansion [64].

In the $c \to \infty$ limit $G^\pm$ become the identity matrix and the ratio of overlaps takes a simple form

$$\frac{\langle \Psi_{\mathrm{BEC}} | \bar{\boldsymbol{\mu}} \rangle}{\langle \Psi_{\mathrm{BEC}} | \bar{\boldsymbol{\lambda}} \rangle} \sqrt{\frac{\langle \bar{\boldsymbol{\lambda}} | \bar{\boldsymbol{\lambda}} \rangle}{\langle \bar{\boldsymbol{\mu}} | \bar{\boldsymbol{\mu}} \rangle}} = \prod_{j=1}^{N/2} \frac{\lambda_j}{\mu_j} + \mathcal{O}(c^{-1}). \tag{51}$$

Next we turn to the $1/c$-expansion of the form factor. It is convenient to introduce some shorthand notations

$$\delta E = E(\bar{\boldsymbol{\mu}}) - E(\bar{\boldsymbol{\lambda}}), \qquad \delta Q_n = Q_n(\bar{\boldsymbol{\mu}}) - Q_n(\bar{\boldsymbol{\lambda}}), \tag{52}$$

where

$$Q_n(\bar{\boldsymbol{\lambda}}) = \sum_{k=1}^N \lambda_k^n. \tag{53}$$

The rapidities $\{\lambda_i\}$ and $\{\mu_j\}$ are solutions to the Bethe equations (7) with Bethe numbers $I_j$ and $J_j$ respectively. The $1/c$-expansion of the rapidity differences $\mu_i - \lambda_i$ is given by

$$\mu_i - \lambda_i = \begin{cases} \frac{2\pi}{L}(J_i - I_i) + \mathcal{O}(c^{-1}), & \text{if } J_i \neq I_i \\ \frac{2\lambda_i \delta E}{c^3 L} + \mathcal{O}(c^{-4}), & \text{otherwise} \end{cases}. \tag{54}$$

The $1/c$-expansion of $V_j^\pm$ is computed by writing

$$V_j^\pm = \exp\left[ \sum_{k=1}^N \log(1 \pm \frac{\mu_k - \lambda_j}{ic}) - \log(1 \pm \frac{\lambda_k - \lambda_j}{ic}) \right], \tag{55}$$

and then Taylor expanding the exponential and the logarithms. For a pair state this gives

$$V_j^+ - V_j^- = \frac{2\lambda_j}{ic^3}\delta E + \frac{i\lambda_j}{c^5}\left[2\delta Q_4 + 4\lambda_j^2\delta E - (\delta E)^2\right] + \mathcal{O}(c^{-6}).\tag{56}$$

Combining (56) and (54) we obtain that the large $c$ limit of the matrix $U$ is given by

$$U_{jk} = -\frac{c^2}{\lambda_j\delta E} + \mathcal{O}(c^0).\tag{57}$$

To evaluate the determinant appearing in the form factor we use that for an invertible matrix $A$ and two vectors $u, v$ we have

$$\det(A + uv^t) = (1 + v^t A^{-1}u)\det A,\tag{58}$$

which implies that

$$\det_{i,j}(\delta_{ij} + U_{ij}) = \mathcal{O}(c^2).\tag{59}$$

Let us now introduce

$$\nu = N - |\{I_i\} \cap \{J_j\}|,\tag{60}$$

i.e. the number of Bethe numbers associated with the rapidities $\bar{\boldsymbol{\lambda}}$ that are distinct from the Bethe numbers corresponding to the rapidities $\bar{\boldsymbol{\mu}}$. Using (54) we find

$$\frac{\prod_{i<j}(\lambda_i - \lambda_j)\prod_{i<j}(\mu_i - \mu_j)}{\prod_{i,j}(\lambda_i - \mu_j)} = \mathcal{O}(c^{3(N-\nu)}).\tag{61}$$

Putting everything together it follows that

$$\frac{\langle\Psi_{\mathrm{BEC}}|\bar{\boldsymbol{\mu}}\rangle\langle\bar{\boldsymbol{\lambda}}|\sigma^2(0)|\bar{\boldsymbol{\mu}}\rangle}{\langle\Psi_{\mathrm{BEC}}|\bar{\boldsymbol{\lambda}}\rangle\langle\bar{\boldsymbol{\mu}}|\bar{\boldsymbol{\mu}}\rangle} = \mathcal{O}(c^{4-3\nu}).\tag{62}$$

This establishes that the $1/c$-expansion of the spectral sum (50) corresponds to an expansion in the number of particle-hole excitations. Since $\nu$ has to be even because of the pair structure of the states, the leading order term for $\bar{\boldsymbol{\mu}} \neq \bar{\boldsymbol{\lambda}}$ is obtained for $\nu = 2$, i.e. two particle-hole excitations and is of order $\mathcal{O}(c^{-2})$. The next terms involve four particle-hole excitations and contribute only at order $\mathcal{O}(c^{-8})$. Since our goal is to compute the relaxation dynamics up to order $c^{-4}$, we can restrict our analysis to two particle-hole excitations.

## 4.3 Two particle-hole excitations

We now fix the rapidities $\boldsymbol{\lambda} > 0$ of the representative state and denote its Bethe numbers by $\{I_j\}$. We then consider $\boldsymbol{\mu} > 0$ such that the corresponding Bethe numbers $J_j$ are equal to $I_j$ except for

$$J_a = I_a + n.\tag{63}$$

The usual exclusion principle in the Bethe ansatz imposes that $n \neq 0$, $J_a > 0$ and $\forall i = 1,...,N$, $J_a \neq I_i$. The Bethe state $|\bar{\boldsymbol{\mu}}\rangle$ constructed in this way is a pair state that corresponds to a two particle-hole excitation over the representative state $|\bar{\boldsymbol{\lambda}}\rangle$. Taking into account only such states in the spectral sum (50) provides a $1/c$-expansion up to and including $\mathcal{O}(c^{-4})$.

Taking the difference between Bethe equations for the roots $\mu_i$ and $\lambda_i$ and using the pair structure we obtain the following expansion for the positive Bethe roots with $i \neq a$

$$\mu_i - \lambda_i = \begin{cases} \frac{2\lambda_i}{L'c^3}\delta E - \frac{2\lambda_i}{c^5 L'}(\delta Q_4 + 2\lambda_i^2\delta E) + \mathcal{O}(c^{-6}), & \text{if } i \neq a \\ \frac{2\pi n}{L'} + \frac{2}{3c^3 L'}\left[N(\mu_a^3 - \lambda_a^3) + 3\lambda_a\delta E + \frac{6\pi n}{L}E(\bar{\boldsymbol{\mu}})\right] + \mathcal{O}(c^{-5}), & \text{if } i = a \,. \end{cases}\tag{64}$$

Here we have introduced the convenient notation

$$L' = L\left(1 + \frac{2\mathcal{D}}{c}\right). \tag{65}$$

We next turn to the $1/c$-expansion of the matrix $U$. We choose $\lambda_p = -\lambda_a$, so that the first term in $U_{jk}$ is $\mathcal{O}(c^{-1})$ except for $j = a$. This gives

$$U_{jk} = \delta_{ja}\frac{i\beta}{V_a^+ - V_a^-}\left[\frac{2c}{c^2 + (\lambda_a - \lambda_k)^2} - \frac{2c}{c^2 + (\lambda_a + \lambda_k)^2}\right]$$
$$+ \frac{2ic}{(V_j^+ - V_j^-)(c^2 + (\lambda_a + \lambda_k)^2)} + \mathcal{O}(c^{-1}), \tag{66}$$

where

$$\beta = \frac{2\pi n}{L'}\left(1 + \frac{\pi n}{\lambda_a L'}\right) = \frac{\delta E}{4\lambda_a} + \mathcal{O}(c^{-3}). \tag{67}$$

We then employ the following identity obtained from (58)

$$\det_{j,k}(\delta_{jk} + m_k\delta_{ja} + u_j v_k) = 1 + m_a + \sum_j u_j v_j + \sum_{j \neq a} u_j\left(v_j m_a - m_j v_a\right), \tag{68}$$

to obtain

$$\det(I + U) = 1 + \frac{i\beta}{V_a^+ - V_a^-}\left[\frac{2}{c} - \frac{2c}{c^2 + 4\lambda_a^2}\right] + if(-\lambda_a) + \frac{\beta}{V_a^+ - V_a^-}\frac{2c}{c^2 + 4\lambda_a^2}f(\lambda_a)$$
$$- \frac{\beta}{V_a^+ - V_a^-}\frac{2}{c}f(-\lambda_a) + \mathcal{O}(c^{-1}). \tag{69}$$

Here we have defined

$$f(z) = \sum_j \frac{1}{V_j^+ - V_j^-}\frac{2c}{c^2 + (z - \lambda_j)^2}. \tag{70}$$

Using that $V_k^+ - V_k^- = -(V_j^+ - V_j^-)$ if $\lambda_k = -\lambda_j$ we have

$$f(z) = \left[\frac{4z}{c^3} - \frac{8z^3}{c^5} + \mathcal{O}(c^{-7})\right]\sum_j \frac{\lambda_j}{V_j^+ - V_j^-} + \left[-\frac{8z}{c^5} + \mathcal{O}(c^{-7})\right]\sum_j \frac{\lambda_j^3}{V_j^+ - V_j^-}. \tag{71}$$

Using (56) we then obtain the following result for the $1/c$-expansion of $f(z)$

$$f(z) = z\frac{2iN}{\delta E}\left[1 + \frac{\frac{\delta Q_4}{\delta E} - \frac{\delta E}{2} - 2z^2}{c^2}\right] + \mathcal{O}(c^{-3}). \tag{72}$$

Noting that

$$\frac{\delta Q_4}{\delta E} = \frac{\delta E}{2} + 2\lambda_a^2 + \mathcal{O}(c^{-1}), \tag{73}$$

we finally arrive at the following expression for the determinant appearing in the form factor

$$\det(I + U) = -\frac{Nc^2}{\lambda_a \delta E} + \mathcal{O}(c^{-1}). \tag{74}$$

The expansion of the remaining terms in the form factor is more straightforward. We find

$$\frac{\prod_{i<j}|\lambda_i - \lambda_j|\prod_{i<j}|\mu_i - \mu_j|}{\prod_{i\neq j}(\lambda_i - \mu_j)} = -4(-1)^{N/2}\frac{|\lambda_a(\lambda_a + \frac{2\pi n}{L'})|}{(2\lambda_a + \frac{2\pi n}{L'})^2} + \mathcal{O}(c^{-3}) \tag{75}$$

$$\sqrt{\prod_{i,j}\frac{\lambda_i - \lambda_j + ic}{\mu_i - \mu_j + ic}} = 1 - \frac{N\delta E}{2c^2} + \mathcal{O}(c^{-3}) \tag{76}$$

$$\frac{V_i^+ - V_i^-}{\mu_i - \lambda_i} = -iL'(1 + \frac{\delta E}{2c^2}) + \mathcal{O}(c^{-3}), \qquad i \neq a, -a \tag{77}$$

$$\delta E = 2\frac{2\pi n}{L'}\left(2\lambda_a + \frac{2\pi n}{L'}\right) + \mathcal{O}(c^{-3})$$

$$\det G(\boldsymbol{\lambda}) = \det G(\boldsymbol{\mu}) = \left(1 + \frac{2\mathcal{D}}{c}\right)^{N-1} + \mathcal{O}(c^{-3}). \tag{78}$$

Putting everything together we obtain

$$\frac{\langle\bar{\boldsymbol{\lambda}}|\sigma^2(0)|\bar{\boldsymbol{\mu}}\rangle}{\sqrt{\langle\bar{\boldsymbol{\mu}}|\bar{\boldsymbol{\mu}}\rangle\langle\bar{\boldsymbol{\lambda}}|\bar{\boldsymbol{\lambda}}\rangle}} = \frac{16|\lambda_a(\lambda_a + \frac{2\pi n}{L'})|}{c^2 L^2(1 + 2\mathcal{D}/c)}\left[1 - \frac{2}{c^2}\left((\tfrac{2\pi n}{L})^2 + \tfrac{4\pi n}{L}\lambda_a + 2\lambda_a^2\right)\right] + \mathcal{O}(c^{-5}). \tag{79}$$

The expansion of the ratio of the normalized overlaps is similarly straightforward

$$\frac{\langle\Psi_{\text{BEC}}|\bar{\boldsymbol{\mu}}\rangle}{\langle\Psi_{\text{BEC}}|\bar{\boldsymbol{\lambda}}\rangle}\sqrt{\frac{\langle\bar{\boldsymbol{\lambda}}|\bar{\boldsymbol{\lambda}}\rangle}{\langle\bar{\boldsymbol{\mu}}|\bar{\boldsymbol{\mu}}\rangle}} = \frac{\lambda_a}{\lambda_a + \frac{2\pi n}{L'}}\left(1 - \frac{2}{c^2}\left(\frac{2\pi n}{L'}\right)^2 - \frac{8\pi n}{L'c^2}\lambda_a\right) + \mathcal{O}(c^{-3}). \tag{80}$$

Our final result for the $1/c$-expansion of the summand in (36) is then

$$\frac{\langle\Psi_{\text{BEC}}|\bar{\boldsymbol{\mu}}\rangle\langle\bar{\boldsymbol{\lambda}}|\sigma^2(0)|\bar{\boldsymbol{\mu}}\rangle}{\langle\Psi_{\text{BEC}}|\bar{\boldsymbol{\lambda}}\rangle\langle\bar{\boldsymbol{\mu}}|\bar{\boldsymbol{\mu}}\rangle} = \frac{16\lambda_a^2}{c^2 L^2(1 + 2\mathcal{D}/c)}\left[1 - \frac{4\left(\lambda_a + \frac{2\pi n}{L}\right)^2}{c^2}\right] + \mathcal{O}(c^{-5}). \tag{81}$$

This is a regular function of $\lambda_a$ and $n$ and in the thermodynamic limit the sums over $\lambda_a$ and $n$ can therefore be turned into integrals

$$\langle\sigma_2(0)\rangle_t = \langle\sigma_2(0)\rangle_\infty \tag{82}$$

$$+ \lim_{\epsilon\to 0}\frac{16}{c^2(1 + 2\frac{\mathcal{D}}{c})}\int_0^\infty \lambda^2(1 - \tfrac{4\mu^2}{c^2})\cos\left(2t(\lambda^2 - \mu^2)\right)\rho(\lambda)\rho_h(\mu)e^{-\epsilon\mu^2}\mathrm{d}\lambda\mathrm{d}\mu$$

$$+ \mathcal{O}(c^{-5}). \tag{83}$$

We refer the reader to Section 3.3 for the $\epsilon \to 0$ limit. Importantly (83) depends on the representative state only via the particle and hole densities. This shows that the typicality assumption underlying (37) indeed holds, at least to the order of the $1/c$-expansion we are working in.

# 5  Calculation of the two-point function $\langle\sigma(x)\sigma(0)\rangle_t$

## 5.1  Spectral representation

The expression (36) for the time evolution obtained within the Quench Action framework is expected to hold for any "weak" operator [53] $\mathcal{O}$, which includes $\sigma(x)\sigma(0)$. Inserting a

resolution of the identity between the two density operators then gives

$$\langle\sigma(x)\sigma(0)\rangle_t = \frac{1}{|\mathfrak{S}_L|}\mathrm{Re}\left[\sum_{\boldsymbol{\lambda}\in\mathfrak{S}_L}\sum_{\boldsymbol{\mu}>0}\sum_{\boldsymbol{\nu}}\frac{\langle\Psi_{\mathrm{BEC}}|\bar{\boldsymbol{\mu}}\rangle\langle\bar{\boldsymbol{\lambda}}|\sigma(0)|\boldsymbol{\nu}\rangle\langle\boldsymbol{\nu}|\sigma(0)|\bar{\boldsymbol{\mu}}\rangle}{\langle\Psi_{\mathrm{BEC}}|\bar{\boldsymbol{\lambda}}\rangle\langle\bar{\boldsymbol{\mu}}|\bar{\boldsymbol{\mu}}\rangle\langle\boldsymbol{\nu}|\boldsymbol{\nu}\rangle}e^{2it(E(\boldsymbol{\lambda})-E(\boldsymbol{\mu}))+ixP(\boldsymbol{\nu})}\right].$$

(84)

We note that the intermediate state $\boldsymbol{\nu}$ does not have to be a pair state. We now proceed as in the case of the one-point by considering a given representative state $\bar{\boldsymbol{\lambda}}$ and defining

$$\mathcal{C}_{\bar{\boldsymbol{\lambda}}}(x,t) = \sum_{\boldsymbol{\mu}>0}\sum_{\boldsymbol{\nu}}\frac{\langle\Psi_{\mathrm{BEC}}|\bar{\boldsymbol{\mu}}\rangle\langle\bar{\boldsymbol{\lambda}}|\sigma(0)|\boldsymbol{\nu}\rangle\langle\boldsymbol{\nu}|\sigma(0)|\bar{\boldsymbol{\mu}}\rangle}{\langle\Psi_{\mathrm{BEC}}|\bar{\boldsymbol{\lambda}}\rangle\langle\bar{\boldsymbol{\mu}}|\bar{\boldsymbol{\mu}}\rangle\langle\boldsymbol{\nu}|\boldsymbol{\nu}\rangle}e^{2it(E(\boldsymbol{\lambda})-E(\boldsymbol{\mu}))+ixP(\boldsymbol{\nu})}.$$

(85)

The usual typicality arguments suggest that in the thermodynamic this quantity will depend on the representative state only via its particle and hole densities. If this holds true then the generalized micro-canonical average in (84) can be dropped and

$$\lim_{L\to\infty}\langle\sigma(x)\sigma(0)\rangle_t = \lim_{L\to\infty}\mathcal{C}_{\bar{\boldsymbol{\lambda}}}(x,t).$$

(86)

We will see below that this is indeed the case due to rather delicate cancellations of contributions that depend on details of the representative state.

The form factors entering (85) are given by [73–78]

$$\frac{\langle\boldsymbol{\mu}|\sigma(0)|\boldsymbol{\lambda}\rangle}{\sqrt{\langle\boldsymbol{\lambda}|\boldsymbol{\lambda}\rangle\langle\boldsymbol{\mu}|\boldsymbol{\mu}\rangle}} = \frac{i^{N+1}(-1)^{N(N-1)/2}(P(\boldsymbol{\lambda})-P(\boldsymbol{\mu}))}{L^N\sqrt{\det G(\boldsymbol{\lambda})\det G(\boldsymbol{\mu})}}\frac{\prod_{i<j}|\lambda_i-\lambda_j|\prod_{i<j}|\mu_i-\mu_j|}{\prod_{i,j}(\mu_j-\lambda_i)}$$
$$\times\sqrt{\prod_{i,j}\frac{\lambda_i-\lambda_j+ic}{\mu_i-\mu_j+ic}}\prod_{j\neq p}(V_j^+-V_j^-)\det_{i,j=1,\dots,N}\left[\delta_{ij}+U_{ij}'\right],$$

(87)

where

$$U_{jk}' = i\frac{\mu_j-\lambda_j}{V_j^+-V_j^-}\left[\frac{2c}{(\lambda_j-\lambda_k)^2+c^2}-\frac{2c}{(\lambda_p-\lambda_k)^2+c^2}\right]\prod_{m\neq j}\frac{\mu_m-\lambda_j}{\lambda_m-\lambda_j}.$$

(88)

## 5.2 Structure of the contributing "excited states"

In order to determine the order $\mathcal{O}(c^{-1})$ in our $1/c$-expansion of (85) we need to know which "excitations" $\boldsymbol{\nu}$ and $\bar{\boldsymbol{\mu}}$ will contribute to the spectral sums. Let us first remark that the limiting value taken by $\langle\sigma(x)\sigma(0)\rangle_t$ when $t\to\infty$ is obtained when $\bar{\boldsymbol{\mu}}=\bar{\boldsymbol{\lambda}}$ in (85), as written in (49). In order to investigate the relaxation dynamics we will thus assume from now on $\bar{\boldsymbol{\mu}}\neq\bar{\boldsymbol{\lambda}}$.

We recall from [64] that the density form-factor for a one particle-hole excitation with rapidities $\boldsymbol{\mu}$ above a state with rapidities $\boldsymbol{\lambda}$ takes the following form at order $\mathcal{O}(c^{-1})$

$$\frac{\langle\boldsymbol{\mu}|\sigma(0)|\boldsymbol{\lambda}\rangle}{\sqrt{\langle\boldsymbol{\lambda}|\boldsymbol{\lambda}\rangle\langle\boldsymbol{\mu}|\boldsymbol{\mu}\rangle}} = \frac{1+\frac{2\mathcal{D}}{c}}{L(1+\frac{2}{cL})}\prod_{i\neq a}\mathrm{sgn}(\lambda_i-\mu_a)\mathrm{sgn}(\lambda_i-\lambda_a)$$
$$\times\left(1+\frac{2(\mu_a-\lambda_a)}{cL}\sum_{i\neq a}\frac{1}{\lambda_i-\mu_a}-\frac{1}{\lambda_i-\lambda_a}\right)+\mathcal{O}(c^{-2}).$$

(89)

The product of signs in this formula arises because we chose an Algebraic Bethe Ansatz description of the eigenstates (8) that is symmetric in the rapidities $\lambda_1,\dots,\lambda_N$, in contrast to the Coordinate Bethe Ansatz description which is antisymmetric. In normalized form and for zero-momentum states, the two are related by a factor $\prod_{i<j}\mathrm{sgn}(\lambda_i-\lambda_j)$ times a phase independent of $\lambda_i$'s [79].

For a two particle-hole excitation where the Bethe numbers of $\boldsymbol{\mu}$ are the same as the ones of $\boldsymbol{\lambda}$ except for $I_a, I_b$, we have [64]

$$
\begin{aligned}
\frac{\langle \boldsymbol{\mu} | \sigma(0) | \boldsymbol{\lambda} \rangle}{\sqrt{\langle \boldsymbol{\lambda} | \boldsymbol{\lambda} \rangle \langle \boldsymbol{\mu} | \boldsymbol{\mu} \rangle}} = & -\prod_{i \neq a,b} \mathrm{sgn}(\lambda_i - \mu_a)\,\mathrm{sgn}(\lambda_i - \lambda_a)\,\mathrm{sgn}(\lambda_i - \mu_b)\,\mathrm{sgn}(\lambda_i - \lambda_b) \\
& \times \mathrm{sgn}(\lambda_a - \lambda_b)\,\mathrm{sgn}(\mu_a - \mu_b) \\
& \times \frac{2}{cL^2} \frac{(\mu_a + \mu_b - \lambda_a - \lambda_b)^2 (\lambda_a - \lambda_b)(\mu_a - \mu_b)}{(\mu_a - \lambda_a)(\mu_b - \lambda_b)(\mu_a - \lambda_b)(\mu_b - \lambda_a)} + \mathcal{O}(c^{-2}).
\end{aligned} \tag{90}
$$

Form factors with a higher number of particle-hole excitations are suppressed by at least a factor $c^{-2}$ and we will ignore them in the following. We are now in a position to identify the dominant "excitations" contributing to the spectral representation at large $c$.

(i) "Type I" configurations contributing at $\mathcal{O}(c^0)$ and higher

Because of the pair structure of both $\bar{\boldsymbol{\lambda}}$ and $\bar{\boldsymbol{\mu}}$ the leading order of the $1/c$-expansion is obtained with states corresponding to a two particle-hole excitation $\bar{\boldsymbol{\mu}}$ above $\bar{\boldsymbol{\lambda}}$ such that the Bethe numbers $I_a, -I_a$ of $\bar{\boldsymbol{\lambda}}$ are replaced by $J_a, -J_a$ in $\bar{\boldsymbol{\mu}}$. We will assume this structure to be satisfied in the following.

Then the intermediate state $\boldsymbol{\nu}$ that provides the leading $\mathcal{O}(c^0)$ contribution is obtained by imposing that it is a one particle-hole excitation above both $\bar{\boldsymbol{\lambda}}$ and $\bar{\boldsymbol{\mu}}$. This implies that the Bethe numbers of $\boldsymbol{\nu}$ have to be the same as those of $\bar{\boldsymbol{\lambda}}$ with the exception of $I_a$ or $-I_a$, which is replaced by either $J_a$ or $-J_a$. These contributions give the full result in the $c \to \infty$ limit, which correspond to a quench directly from the BEC to the Tonks-Girardeau gas [45]. However, they also incorporate $c^{-1}$ corrections due to subleading terms in the form factors.

(ii) "Type II" configurations contributing at $\mathcal{O}(c^{-1})$

At order $\mathcal{O}(c^{-1})$ contribution arise from other terms in the spectral sum as well. One class of terms corresponds to the case where $\boldsymbol{\nu}$ is equal to $\bar{\boldsymbol{\lambda}}$ ($\bar{\boldsymbol{\mu}}$) and corresponds to a two particle-hole excitations above $\bar{\boldsymbol{\mu}}$ ($\bar{\boldsymbol{\lambda}}$). In this case one of the two form factors in (85) reduces to the expectation value of $\sigma$ which equals the density $\mathcal{D}$, while other form factor is of order $\mathcal{O}(c^{-1})$ since it involves states related by two particle-hole excitations. Closer inspection of (85) reveals that these contributions cancel

$$
\mathcal{D}\langle \sigma(0) \rangle_t - \mathcal{D}^2 + \mathcal{D}\langle \sigma(x) \rangle_t - \mathcal{D}^2 = 0. \tag{91}
$$

Here we have used that since $\sigma$ is a conserved quantity we have

$$
\langle \sigma(0) \rangle_t = \langle \sigma(0) \rangle_0 = \mathcal{D}. \tag{92}
$$

This leaves one remaining source for $\mathcal{O}(c^{-1})$ contributions, namely when the $\boldsymbol{\nu}$ correspond to a one particle-hole excitation above $\bar{\boldsymbol{\lambda}}$ ($\bar{\boldsymbol{\mu}}$) and a two particle-hole excitation above $\bar{\boldsymbol{\mu}}$ ($\bar{\boldsymbol{\lambda}}$). As we will see below these terms give non-vanishing contributions to the spectral sum.

## 5.3 Contributions arising from type I configurations

We now consider case (i) above, in which $\boldsymbol{\lambda} \neq \boldsymbol{\mu}$ and $\boldsymbol{\nu}$ corresponds to a one particle-hole excitation above both $\bar{\boldsymbol{\lambda}}$ and $\bar{\boldsymbol{\mu}}$. We denote the corresponding contribution to (85) by $\mathcal{C}_{\bar{\boldsymbol{\lambda}}}^{1,1}(x,t)$. These contributions are sketched in Figure 5. The four possible choices for $\boldsymbol{\nu}$ can be accounted for by replacing the rapidity $\lambda_a$ by $\mu_a$ in $\boldsymbol{\nu}$, but allowing both $\lambda_a$ and $\mu_a$ to take values between

$$|\bar{\boldsymbol{\lambda}}\rangle \quad \cdot \quad \cdot \quad \cdot -I_a \cdot \quad \cdot \quad \cdot \quad I_a \quad \cdot \quad \cdot \quad \cdot$$

$$|\boldsymbol{\nu}\rangle \quad \cdot \quad \cdot \quad \cdot -J_a \cdot \quad \cdot \quad \cdot \quad I_a \quad \cdot \quad \cdot \quad \cdot$$

$$|\bar{\boldsymbol{\mu}}\rangle \quad \cdot \quad \cdot \quad \cdot -J_a \cdot \quad \cdot \quad \cdot \quad J_a \quad \cdot \quad \cdot \quad \cdot$$

Figure 5: An example of a type I excitation. Dots indicate Bethe numbers that are the same. We see that $\boldsymbol{\nu}$ differs from $\bar{\boldsymbol{\lambda}}$ by the replacement $-I_a \to -J_a$, while $\bar{\boldsymbol{\mu}}$ differs from $\boldsymbol{\nu}$ by replacing $I_a \to J_a$.

$-\infty$ and $\infty$. At order $\mathcal{O}(c^{-1})$ the form factors entering the spectral sum are given by (89), and the overlaps are

$$\frac{\langle \Psi_{\text{BEC}} | \bar{\boldsymbol{\mu}}\rangle}{\langle \Psi_{\text{BEC}} | \bar{\boldsymbol{\lambda}}\rangle} \sqrt{\frac{\langle \bar{\boldsymbol{\lambda}} | \bar{\boldsymbol{\lambda}}\rangle}{\langle \bar{\boldsymbol{\mu}} | \bar{\boldsymbol{\mu}}\rangle}} = \left| \frac{\lambda_a}{\mu_a} \right| + \mathcal{O}(c^{-2}). \tag{93}$$

Here the absolute values arise because in (38) the $\lambda_j$ denote by definition the positive rapidities in $\bar{\boldsymbol{\lambda}}$ only, whereas $\lambda_a, \mu_a$ can be either positive or negative. The signs appearing in the form factor (89) have to be treated carefully and give rise to a factor $\text{sgn}(\lambda_a \mu_a)$ in the summand in (85). The $1/c$-expansion of this summand reads

$$\frac{\langle \Psi_{\text{BEC}} | \bar{\boldsymbol{\mu}}\rangle \langle \bar{\boldsymbol{\lambda}} | \sigma(0) | \boldsymbol{\nu}\rangle \langle \boldsymbol{\nu} | \sigma(0) | \bar{\boldsymbol{\mu}}\rangle}{\langle \Psi_{\text{BEC}} | \bar{\boldsymbol{\lambda}}\rangle \langle \bar{\boldsymbol{\mu}} | \bar{\boldsymbol{\mu}}\rangle \langle \boldsymbol{\nu} | \boldsymbol{\nu}\rangle} = \frac{(1 + 2\mathcal{D}/c)^2}{L^2 (1 + 2/(cL))^2} \frac{\lambda_a}{\mu_a}$$

$$\times \left( 1 + \frac{4(\mu_a - \lambda_a)}{cL} \sum_{\substack{i \\ \lambda_i \neq \pm \lambda_a}} \frac{1}{\lambda_i - \mu_a} - \frac{1}{\lambda_i - \lambda_a} + \frac{\mu_a - \lambda_a}{cL} \left[ \frac{1}{\lambda_a} - \frac{1}{\mu_a} \right] \right) + \mathcal{O}(c^{-2}). \tag{94}$$

This allows us to cast the corresponding contribution to (85) in the form

$$C_{\bar{\boldsymbol{\lambda}}}^{1,1}(x,t) = (1 + 2\mathcal{D}/c)^2 \sum_{a=0}^{3} \Sigma_a(x',t), \tag{95}$$

where

$$\Sigma_0(x',t) = \frac{1}{L^2} \sum_{\lambda_a \in \Lambda} \sum_{\substack{\mu_a \\ \mu_a \notin \Lambda}} \frac{\lambda_a}{\mu_a} e^{2it(\lambda_a^2 - \mu_a^2) + ix'(\mu_a - \lambda_a)}, \tag{96}$$

$$\Sigma_1(x',t) = \frac{1}{L^3} \sum_{\lambda_a \in \Lambda} \sum_{\substack{\mu_a \\ \mu_a \notin \Lambda}} \sum_{\substack{\lambda_i \in \Lambda \\ \lambda_i \neq \lambda_a, -\lambda_a}} \frac{g(\lambda_a, \mu_a)}{\mu_a(\lambda_i - \mu_a)}, \tag{97}$$

$$\Sigma_2(x',t) = \frac{1}{L^3} \sum_{\lambda_a \in \Lambda} \sum_{\substack{\mu_a \\ \mu_a \notin \Lambda}} \sum_{\substack{\lambda_i \in \Lambda \\ \lambda_i \neq \lambda_a, -\lambda_a}} \frac{g(\lambda_a, \mu_a)}{\mu_a(\lambda_i - \lambda_a)}, \tag{98}$$

$$\Sigma_3(x',t) = \frac{1}{L^3} \sum_{\lambda_a \in \Lambda} \sum_{\substack{\mu_a \\ \mu_a \notin \Lambda}} \frac{\lambda_a(\mu_a - \lambda_a)}{\mu_a} \left[ \frac{1}{\lambda_a} - \frac{1}{\mu_a} \right] e^{2it(\lambda_a^2 - \mu_a^2) + ix'(\mu_a - \lambda_a)}. \tag{99}$$

Here we introduced a set $\Lambda = \{\lambda_i | i = 1, ..., N/2\} \cup \{-\lambda_i | i = 1, ..., N/2\}$ and defined

$$g(\lambda, \mu) = 4\lambda(\mu - \lambda) e^{2it(\lambda^2 - \mu^2) + ix'(\mu - \lambda)}. \tag{100}$$

We recall that $x'$ was defined in (23). It appears here since at order $\mathcal{O}(c^{-1})$ one has $E(\boldsymbol{\lambda}) - E(\boldsymbol{\mu}) = \lambda_a^2 - \mu_a^2 + \mathcal{O}(c^{-2})$ and $P(\boldsymbol{\nu}) = x'(\mu_a - \lambda_a) + \mathcal{O}(c^{-2})$.

The contribution $\Sigma_0(x, t)$ can be straightforwardly turned into a principal part integral in the thermodynamic limit, while the remaining sums can be carried out in the thermodynamic limit using the following Lemmas.

**Lemma 1.** *Let $f(\lambda, \mu, \nu)$ be a regular function that grows sufficiently slowly at infinity. Then in the thermodynamic limit we obtain*

$$\frac{1}{L^3} \sum_{\lambda_a \in \Lambda} \sum_{\lambda_i \in \Lambda} \sum_{\substack{\lambda_j \in \Lambda \\ \lambda_j \neq \lambda_i}} \frac{f(\lambda_a, \lambda_j, \lambda_i)}{\lambda_i(\lambda_j - \lambda_i)} = \int_{-\infty}^{\infty} d\lambda \rho(\lambda) \fint d\nu \rho(\nu) \fint d\mu \rho(\mu) \frac{f(\lambda, \mu, \nu)}{\nu(\mu - \nu)}$$

$$+ \frac{\pi^2 \rho(0)^2 - \Omega(\boldsymbol{\lambda})}{2} \int_{-\infty}^{\infty} f(\lambda, 0, 0) \rho(\lambda) d\lambda + \mathcal{O}(L^{-1}). \quad (101)$$

Here we have defined

$$\Omega(\boldsymbol{\lambda}) \equiv \lim_{L \to \infty} \frac{1}{L^2} \sum_{\lambda \in \Lambda} \frac{1}{\lambda^2}. \quad (102)$$

We stress that $\Omega(\boldsymbol{\lambda})$ is a quantity that in the thermodynamic limit depends on the choice of representative state not only through the root density $\rho(\lambda)$. A proof of Lemma 1 is given in Appendix B.

**Lemma 2.** *Let $f(\lambda, \mu, \nu)$ be a regular function that grows sufficiently slowly at infinity. Then in the thermodynamic limit*

$$\frac{1}{L^3} \sum_{\lambda_a \in \Lambda} \sum_{n \neq 0} \sum_{\lambda_i \in \Lambda} \frac{f(\lambda_a, \lambda_i + \frac{2\pi n}{L}, \lambda_i)}{(\lambda_i + \frac{2\pi n}{L})(-\frac{2\pi n}{L})} = \frac{1}{2\pi} \int_{-\infty}^{\infty} d\lambda \rho(\lambda) \fint d\nu \rho(\nu) \fint d\mu \frac{f(\lambda, \mu, \nu)}{\mu(\nu - \mu)}$$

$$+ \frac{1}{2\pi} \rho(0) \pi^2 \int_{-\infty}^{\infty} f(\lambda, 0, 0) \rho(\lambda) d\lambda - \Omega(\boldsymbol{\lambda}) \int_{-\infty}^{\infty} f(\lambda, 0, 0) \rho(\lambda) d\lambda + \mathcal{O}(L^{-1}).$$

$$(103)$$

A proof of Lemma 1 is given in Appendix C.

### 5.3.1 First sum $\Sigma_1(x, t)$

Writing out the various constraints in the summations explicitly we have

$$\Sigma_1(x, t) = \frac{1}{L^3} \sum_{\lambda_a \in \Lambda} \sum_{n \neq 0} \sum_{\lambda_i \in \Lambda} \frac{g(\lambda_a, \lambda_i + \frac{2\pi n}{L})}{(\lambda_i + \frac{2\pi n}{L})(-\frac{2\pi n}{L})} - \frac{1}{L^3} \sum_{\lambda_a \in \Lambda} \sum_{\lambda_j \in \Lambda} \sum_{\substack{\lambda_i \in \Lambda \\ i \neq j}} \frac{g(\lambda_a, \lambda_j)}{\lambda_j(\lambda_i - \lambda_j)}$$

$$- \frac{1}{L^3} \sum_{\lambda_a \in \Lambda} \sum_{\substack{\mu_a \\ \mu_a \notin \Lambda}} \frac{g(\lambda_a, \mu_a)}{\mu_a(\lambda_a - \mu_a)} - \frac{1}{L^3} \sum_{\lambda_a \in \Lambda} \sum_{\substack{\mu_a \\ \mu_a \notin \Lambda}} \frac{g(\lambda_a, \mu_a)}{\mu_a(-\lambda_a - \mu_a)}. \quad (104)$$

We note that since the states have a pair structure, $N$ is even and the Bethe numbers are half-odd integers, and so neither $\mu_a$ or $\lambda_i + \frac{2\pi n}{L}$ can vanish in the denominators.

The last two sums are two-dimensional sums with a prefactor $1/L^3$ but only simple poles and hence vanish in the thermodynamic limit. The remaining two sums in (104) can be carried out using Lemma 2 and Lemma 1 respectively. This gives

$$\Sigma_1(x', t) = -4 \int_{-\infty}^{\infty} d\lambda \rho(\lambda) \fint d\mu \rho_h(\mu) \frac{\lambda}{\mu} (\mu - \lambda) \tilde{\rho}(\mu) e^{2it(\lambda^2 - \mu^2) + ix'(\mu - \lambda)}$$

$$+ \left[ 2\Omega(\boldsymbol{\lambda}) - 4\pi^2 \rho(0) \left( \frac{1}{2\pi} - \frac{\rho(0)}{2} \right) \right] \int_{-\infty}^{\infty} \lambda^2 \rho(\lambda) e^{2it\lambda^2 - ix'\lambda} d\lambda, \quad (105)$$

where $\tilde{\rho}(\lambda)$ denotes the Hilbert transform of $\rho(\lambda)$ defined by

$$\tilde{\rho}(\lambda) = \int \frac{\rho(\nu)}{\lambda - \nu} d\nu. \tag{106}$$

### 5.3.2 Second sum $\Sigma_2(x', t)$

Writing out the constraints on the various summations explicitly we have

$$\Sigma_2(x', t) = \frac{1}{L^3} \sum_{\lambda_a \in \Lambda} \sum_{n} \sum_{\substack{\lambda_i \in \Lambda \\ \lambda_i \neq \lambda_a}} \frac{g(\lambda_a, \frac{2\pi(n+1/2)}{L})}{\frac{2\pi(n+1/2)}{L}(\lambda_i - \lambda_a)} - \frac{1}{L^3} \sum_{\lambda_a \in \Lambda} \sum_{\lambda_j \in \Lambda} \sum_{\substack{\lambda_i \in \Lambda \\ i \neq a}} \frac{g(\lambda_a, \lambda_j)}{\lambda_j(\lambda_i - \lambda_a)}$$
$$+ \frac{1}{2L^3} \sum_{\lambda_a \in \Lambda} \sum_{\substack{\mu_a \\ \mu_a \notin \Lambda}} \frac{g(\lambda_a, \mu_a)}{\mu_a \lambda_a}. \tag{107}$$

The third sum is a two-dimensional sum with a prefactor $1/L^3$ and no double poles and hence vanishes in the thermodynamic limit. The first two sums can be turned into principal value integrals, which gives

$$\Sigma_2(x', t) = -4 \int_{-\infty}^{\infty} d\lambda \rho(\lambda) \int d\mu \rho_h(\mu) \frac{\lambda}{\mu}(\mu - \lambda)\tilde{\rho}(\lambda) e^{2it(\lambda^2 - \mu^2) + ix'(\mu - \lambda)}. \tag{108}$$

### 5.3.3 Third sum $\Sigma_3(x', t)$

The third sum is a two-dimensional sum with a prefactor $1/L^3$, so can contribute in the thermodynamic limit only if there is a double pole. It follows that

$$\Sigma_3(x', t) = \frac{1}{L^3} \sum_{\lambda_a \in \Lambda} \sum_{\mu_a \notin \Lambda} \frac{\lambda_a^2}{\mu_a^2} e^{2it(\lambda_a^2 - \mu_a^2) + ix'(\mu_a - \lambda_a)} + \mathcal{O}(L^{-1}). \tag{109}$$

By writing out the constraint explicitly we have

$$\Sigma_3(x', t) = \frac{1}{L^3} \sum_{\lambda_a \in \Lambda} \sum_{n} \frac{\lambda_a^2}{(\frac{2\pi(n+1/2)}{L})^2} e^{2it(\lambda_a^2 - (\frac{2\pi(n+1/2)}{L})^2) + ix'(\frac{2\pi(n+1/2)}{L} - \lambda_a)}$$
$$- \frac{1}{L^3} \sum_{\lambda_a \in \Lambda} \sum_{\lambda_i \in \Lambda} \frac{\lambda_a^2}{\lambda_i^2} e^{2it(\lambda_a^2 - \lambda_i^2) + ix'(\lambda_i - \lambda_a)}. \tag{110}$$

We see that the sum over $\lambda_a$ can be turned into an integral, while the remaining sums can be respectively carried out explicitly and expressed in terms of $\Omega(\lambda)$ (102). This gives

$$\Sigma_3(x', t) = \left[\frac{1}{4} - \Omega(\lambda)\right] \int_{-\infty}^{\infty} \lambda^2 \rho(\lambda) e^{2it\lambda^2 - ix'\lambda} d\lambda + \mathcal{O}(L^{-1}). \tag{111}$$

### 5.3.4 Result

Putting everything together, we obtain the following result for the contribution of two one particle-hole excitations to the spectral sum (85)

$$\mathcal{C}_{\tilde{\lambda}}^{1,1}(x, t) =$$
$$(1 + 2\mathcal{D}/c)^2 \int_{-\infty}^{\infty} d\lambda \rho(\lambda) \int d\mu \rho_h(\mu) \frac{\lambda}{\mu} \left(1 - \frac{4}{c}(\mu - \lambda)(\tilde{\rho}(\mu) - \tilde{\rho}(\lambda))\right) e^{2it(\lambda^2 - \mu^2) + ix'(\mu - \lambda)}$$
$$+ \frac{1}{c} \left[\frac{1}{4} - 2\pi\rho(0) + 2\pi^2 \rho(0)^2 + \Omega(\lambda)\right] \int_{-\infty}^{\infty} \lambda^2 \rho(\lambda) e^{2it\lambda^2 - ix'\lambda} d\lambda. \tag{112}$$

We stress that $C_{\bar{\lambda}}^{1,1}(x,t)$ depends on the representative state $\boldsymbol{\lambda}$ not only through the root density $\rho$, but via the quantity $\Omega(\boldsymbol{\lambda})$ (102) as well.

## 5.4 Contributions arising from type II configurations

Let us denote by $C_{\bar{\lambda}}^{2,1}(x,t)$ the sum of contributions of type-II configurations to (85), i.e. configurations where $\boldsymbol{\nu}$ corresponds to a one particle-hole excitation above $\bar{\boldsymbol{\lambda}}$ ($\bar{\boldsymbol{\mu}}$) and a two particle-hole excitation above $\bar{\boldsymbol{\mu}}$ ($\bar{\boldsymbol{\lambda}}$) respectively . There are altogether four cases:

(i) The Bethe numbers of $\boldsymbol{\nu}$ are those of $\bar{\boldsymbol{\lambda}}$ except for the replacement of $I_a$ or $-I_a$ by $K_a$. Denoting the corresponding root by $\nu_a$ we have the following restrictions: $\forall i, \mu_a \neq \lambda_i$; $\forall i, \nu_a \neq \lambda_i$; $\nu_a \neq \mu_a, -\mu_a$.

(ii) The Bethe numbers of $\boldsymbol{\nu}$ are those of $\bar{\boldsymbol{\mu}}$ with only $J_a$ or $-J_a$ replaced by a $K_a$. Denoting $\nu_a$ the corresponding root, we have the restrictions $\forall i, \mu_a \neq \lambda_i$ and $\forall i, \nu_a \neq \lambda_i$ and $\nu_a \neq \mu_a, -\mu_a$. Cases (i) and (ii) are sketched in Figure 6.

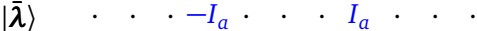

Figure 6: Cases (i) and (ii) of type II excitations.

(iii) The Bethe numbers of $\boldsymbol{\nu}$ are those of $\bar{\boldsymbol{\lambda}}$ with only $I_b$ or $-I_b$ ($b \neq a$) replaced by $J_a$ or $-J_a$. The restrictions on the rapidities are $\lambda_b \neq \lambda_a, -\lambda_a$; $\forall i, \mu_a \neq \lambda_i$.

(iv) The Bethe numbers of $\boldsymbol{\nu}$ are those of $\bar{\boldsymbol{\mu}}$ with only $I_b$ or $-I_b$ ($b \neq a$) replaced by $I_a$ or $-I_a$. The restrictions on the rapidities are $\lambda_b \neq \lambda_a, -\lambda_a$; $\forall i, \mu_a \neq \lambda_i$. Cases (iii) and (iv) are sketched in Figure 7.

Figure 7: Cases (iii) and (iv) of type II excitations.

Case (i) can be accounted for by always changing $\lambda_a$ for $\nu_a$, but allowing $\lambda_a$ to range between $-\infty$ and $\infty$. One can also allow $\mu_a$ to range between $-\infty$ and $\infty$ by introducing a combinatorial factor $\frac{1}{2}$. In case (ii) the same holds true with $\lambda_a$ and $\mu_a$ interchanged. Case (iii) can be accounted for by always changing $\lambda_b$ for $\mu_a$, but allowing both $\lambda_b$ and $\mu_a$ to range between $-\infty$ and $\infty$. One can also allow $\lambda_a$ to range between $-\infty$ and $\infty$ by introducing a combinatorial factor $\frac{1}{2}$. In case (iv) the same holds true with $\lambda_a$ and $\mu_a$ interchanged.

In cases (i) and (ii) the product of all the signs appearing in (89) and (90) give a factor $-\text{sgn}(\lambda_a \mu_a)$. In cases (iii) and (iv) they give a factor $\text{sgn}(\lambda_a \mu_a)$. It follows that in these four

cases we have

$$
\frac{\langle \Psi_{\text{BEC}} | \bar{\boldsymbol{\mu}} \rangle \langle \bar{\boldsymbol{\lambda}} | \sigma(0) | \boldsymbol{\nu} \rangle \langle \boldsymbol{\nu} | \sigma(0) | \bar{\boldsymbol{\mu}} \rangle}{\langle \Psi_{\text{BEC}} | \bar{\boldsymbol{\lambda}} \rangle \langle \bar{\boldsymbol{\mu}} | \bar{\boldsymbol{\mu}} \rangle \langle \boldsymbol{\nu} | \boldsymbol{\nu} \rangle} e^{2it(E(\boldsymbol{\lambda}) - E(\boldsymbol{\mu})) + ix' P(\boldsymbol{\nu})}
$$

$$
= \begin{cases}
-\frac{2}{cL^3} \frac{(\nu_a - \lambda_a)^2 2\lambda_a(\nu_a + \lambda_a)}{(\nu_a^2 - \mu_a^2)(\lambda_a^2 - \mu_a^2)} e^{2it(\lambda_a^2 - \mu_a^2)} e^{ix'(\nu_a - \lambda_a)} & \text{case (i)} \\
-\frac{2}{cL^3} \frac{(\nu_a - \mu_a)^2 2\lambda_a^2(\nu_a + \mu_a)}{(\nu_a^2 - \lambda_a^2)(\mu_a^2 - \lambda_a^2)\mu_a} e^{2it(\lambda_a^2 - \mu_a^2)} e^{ix'(\nu_a - \mu_a)} & \text{case (ii)} \\
\frac{2}{cL^3} \frac{(\lambda_b - \mu_a)^2 2\lambda_a^2(\mu_a + \lambda_b)}{(\lambda_b^2 - \lambda_a^2)(\mu_a^2 - \lambda_a^2)\mu_a} e^{2it(\lambda_a^2 - \mu_a^2)} e^{ix'(\mu_a - \lambda_b)} & \text{case (iii)} \\
\frac{2}{cL^3} \frac{(\lambda_b - \lambda_a)^2 2\lambda_a(\lambda_b + \lambda_a)}{(\lambda_b^2 - \mu_a^2)(\lambda_a^2 - \mu_a^2)} e^{2it(\lambda_a^2 - \mu_a^2)} e^{ix'(\lambda_a - \lambda_b)} & \text{case (iv)}
\end{cases} + \mathcal{O}(c^{-2}). \tag{113}
$$

In order to proceed it is convenient to decompose the rational functions in (113) using

$$
\frac{(\nu_a - \lambda_a)^2 2\lambda_a(\nu_a + \lambda_a)}{(\nu_a^2 - \mu_a^2)(\lambda_a^2 - \mu_a^2)} = \frac{\nu_a - \lambda_a}{\lambda_a + \mu_a} + \frac{\lambda_a - \nu_a}{\mu_a - \lambda_a} + \frac{\lambda_a(\nu_a - \lambda_a)}{\nu_a(\mu_a - \nu_a)} + \frac{\lambda_a(\lambda_a - \nu_a)}{\nu_a(\nu_a + \mu_a)} ,
$$

$$
\frac{(\nu_a - \mu_a)^2 2\lambda_a^2(\nu_a + \mu_a)}{(\nu_a^2 - \lambda_a^2)(\mu_a^2 - \lambda_a^2)\mu_a} = \frac{\nu_a - \mu_a}{\mu_a + \lambda_a} + \frac{\mu_a - \nu_a}{\lambda_a - \mu_a} + \frac{\nu_a(\nu_a - \mu_a)}{\mu_a(\lambda_a - \nu_a)} + \frac{\nu_a(\mu_a - \nu_a)}{\mu_a(\lambda_a + \nu_a)} ,
$$

$$
\frac{(\lambda_b - \mu_a)^2 2\lambda_a^2(\mu_a + \lambda_b)}{(\lambda_b^2 - \lambda_a^2)(\mu_a^2 - \lambda_a^2)\mu_a} = \frac{\mu_a - \lambda_b}{\lambda_a - \mu_a} + \frac{\lambda_b - \mu_a}{\lambda_a + \mu_a} + \frac{\lambda_b(\lambda_b - \mu_a)}{\mu_a(\lambda_a - \lambda_b)} + \frac{\lambda_b(\mu_a - \lambda_b)}{\mu_a(\lambda_a + \lambda_b)} ,
$$

$$
\frac{(\lambda_b - \lambda_a)^2 2\lambda_a(\lambda_b + \lambda_a)}{(\lambda_b^2 - \mu_a^2)(\lambda_a^2 - \mu_a^2)} = \frac{\lambda_a - \lambda_b}{\mu_a - \lambda_a} + \frac{\lambda_b - \lambda_a}{\lambda_a + \mu_a} + \frac{\lambda_a(\lambda_b - \lambda_a)}{\lambda_b(\mu_a - \lambda_b)} + \frac{\lambda_a(\lambda_a - \lambda_b)}{\lambda_b(\lambda_b + \mu_a)} . \tag{114}
$$

Using (113) and (114) we can express the sum of all type-II contributions to (85) in the form

$$
\mathcal{C}_{\bar{\lambda}}^{2,1}(x, t) = \frac{2}{c} \Big[ -\Sigma_1'(x', t) - \Sigma_2'(x', t) + \Sigma_3'(x', t) + \Sigma_4'(x', t) \Big], \tag{115}
$$

where

$$
\Sigma_1'(x', t) = \frac{1}{L^3} \sum_{\lambda_a \in \Lambda} \sum_{\substack{\mu_a \\ \mu_a \notin \Lambda}} \sum_{\substack{\nu_a \\ \nu_a \notin \Lambda \\ \nu_a \neq \mu_a, -\mu_a}} \left[ \frac{\nu_a - \lambda_a}{\lambda_a + \mu_a} + \frac{\lambda_a(\nu_a - \lambda_a)}{\nu_a(\mu_a - \nu_a)} \right] e^{2it(\lambda_a^2 - \mu_a^2)} e^{ix'(\nu_a - \lambda_a)} ,
$$

$$
\Sigma_2'(x', t) = \frac{1}{L^3} \sum_{\lambda_a \in \Lambda} \sum_{\substack{\mu_a \\ \mu_a \notin \Lambda}} \sum_{\substack{\nu_a \\ \nu_a \notin \Lambda \\ \nu_a \neq \mu_a, -\mu_a}} \left[ \frac{\nu_a - \mu_a}{\mu_a + \lambda_a} + \frac{\nu_a(\nu_a - \mu_a)}{\mu_a(\lambda_a - \nu_a)} \right] e^{2it(\lambda_a^2 - \mu_a^2)} e^{ix'(\nu_a - \mu_a)} ,
$$

$$
\Sigma_3'(x', t) = \frac{1}{L^3} \sum_{\lambda_a \in \Lambda} \sum_{\substack{\mu_a \\ \mu_a \notin \Lambda}} \sum_{\substack{\lambda_b \in \Lambda \\ \lambda_b \neq \lambda_a, -\lambda_a}} \left[ \frac{\mu_a - \lambda_b}{\lambda_a - \mu_a} + \frac{\lambda_b(\lambda_b - \mu_a)}{\mu_a(\lambda_a - \lambda_b)} \right] e^{2it(\lambda_a^2 - \mu_a^2)} e^{ix'(\mu_a - \lambda_b)} ,
$$

$$
\Sigma_4'(x', t) = \frac{1}{L^3} \sum_{\lambda_a \in \Lambda} \sum_{\substack{\mu_a \\ \mu_a \notin \Lambda}} \sum_{\substack{\lambda_b \in \Lambda \\ \lambda_b \neq \lambda_a, -\lambda_a}} \left[ \frac{\lambda_a - \lambda_b}{\mu_a - \lambda_a} + \frac{\lambda_a(\lambda_b - \lambda_a)}{\lambda_b(\mu_a - \lambda_b)} \right] e^{2it(\lambda_a^2 - \mu_a^2)} e^{ix'(\lambda_a - \lambda_b)} . \tag{116}
$$

In all four contributions $\Sigma_j'(x', t)$ the respective first term only involves simple poles and therefore can be straightforwardly expressed in terms of principal value integrals in the thermodynamic limit. The other terms involve two simple poles and require a more elaborate treatment.

### 5.4.1  First term $\Sigma_1'(x', t)$

The contribution to $\Sigma_1'(x', t)$ involving two simple poles is of the form

$$
S_L[f] = \frac{1}{L^3} \sum_{\lambda_a \in \Lambda} \sum_{\mu_a \notin \Lambda} \sum_{\substack{\nu_a \notin \Lambda \\ \nu_a \neq \pm \mu_a}} \frac{f(\lambda_a, \mu_a, \nu_a)}{\nu_a(\mu_a - \nu_a)} , \tag{117}
$$

where

$$f(\lambda, \mu, \nu) = \lambda(\nu - \lambda)e^{2it(\lambda^2 - \mu^2) + ix'(\nu - \lambda)}. \tag{118}$$

Resolving all the constraints, we have at leading order in $1/c$

$$
\begin{aligned}
S_L[f] &= \frac{1}{L^3} \sum_{\lambda_a \in \Lambda} \sum_m \sum_{n \neq 0} \frac{f(\lambda_a, \lambda_a + \frac{2\pi m}{L}, \lambda_a + \frac{2\pi(n+m)}{L})}{(\lambda_a + \frac{2\pi(n+m)}{L})(-\frac{2\pi n}{L})} \\
&- \frac{1}{L^3} \sum_{\lambda_a \in \Lambda} \sum_{\lambda_i \in \Lambda} \sum_{m \neq 0} \frac{f(\lambda_a, \lambda_i + \frac{2\pi m}{L}, \lambda_i)}{\lambda_i \frac{2\pi m}{L}} - \frac{1}{L^3} \sum_{\lambda_a \in \Lambda} \sum_{\lambda_j \in \Lambda} \sum_{n \neq 0} \frac{f(\lambda_a, \lambda_j, \lambda_j + \frac{2\pi n}{L})}{(\lambda_j + \frac{2\pi n}{L})(-\frac{2\pi n}{L})} \\
&+ \frac{1}{L^3} \sum_{\lambda_a \in \Lambda} \sum_{\lambda_i \in \Lambda} \sum_{\substack{\lambda_j \in \Lambda \\ \lambda_j \neq \lambda_i}} \frac{f(\lambda_a, \lambda_j, \lambda_i)}{\lambda_i(\lambda_j - \lambda_i)} \\
&+ \frac{1}{2L^3} \sum_{\lambda_a \in \Lambda} \sum_{n \neq 0} \frac{f(\lambda_a, \frac{2\pi(n+1/2)}{L}, -\frac{2\pi(n+1/2)}{L})}{(\frac{2\pi(n+1/2)}{L})^2} - \frac{1}{2L^3} \sum_{\lambda_a \in \Lambda} \sum_{\lambda_i \in \Lambda} \frac{f(\lambda_a, \lambda_i, -\lambda_i)}{\lambda_i^2}. \tag{119}
\end{aligned}
$$

The first two contributions can be computed by first summing over $m$, and then summing over $n$ and $\lambda_i$ respectively, which involves one-dimensional sums with only a single simple pole. In the thermodynamic limit they can be readily turned into principal value integrals. The fifth and sixth terms are double sums with a factor $L^{-3}$ and hence are completely dominated by their respective double poles. They yield

$$
\begin{aligned}
&\frac{1}{2L^3} \sum_{\lambda_a \in \Lambda} \sum_{n \neq 0} \frac{f(\lambda_a, \frac{2\pi(n+1/2)}{L}, -\frac{2\pi(n+1/2)}{L})}{(\frac{2\pi(n+1/2)}{L})^2} - \frac{1}{2L^3} \sum_{\lambda_a \in \Lambda} \sum_{\lambda_i \in \Lambda} \frac{f(\lambda_a, \lambda_i, -\lambda_i)}{\lambda_i^2} \\
&= \left[\frac{1}{8} - \frac{\Omega(\lambda)}{2}\right] \int_{-\infty}^{\infty} f(\lambda, 0, 0)\rho(\lambda)\mathrm{d}\lambda + \mathcal{O}(L^{-1}). \tag{120}
\end{aligned}
$$

The third term is of a very similar structure to Lemma 2 (103) and can be treated analogously. We write

$$
\begin{aligned}
\frac{1}{L^3} \sum_{\lambda_a \in \Lambda} \sum_{\lambda_j \in \Lambda} \sum_{n \neq 0} \frac{f(\lambda_a, \lambda_j, \lambda_j + \frac{2\pi n}{L})}{(\lambda_j + \frac{2\pi n}{L})(-\frac{2\pi n}{L})} &= \frac{1}{L^3} \sum_{\lambda_a \in \Lambda} \sum_{\lambda_j \in \Lambda} \frac{1}{\lambda_j} \sum_n \frac{f(\lambda_a, \lambda_j, \lambda_j + \frac{2\pi n}{L})}{\lambda_j + \frac{2\pi n}{L}} \\
&- \frac{1}{L^3} \sum_{\lambda_a \in \Lambda} \sum_{\lambda_j \in \Lambda} \frac{1}{\lambda_j} \sum_{n \neq 0} \frac{f(\lambda_a, \lambda_j, \lambda_j + \frac{2\pi n}{L})}{\frac{2\pi n}{L}} - \frac{1}{L^3} \sum_{\lambda_a \in \Lambda} \sum_{\lambda_j \in \Lambda} \frac{f(\lambda_a, \lambda_j, \lambda_j)}{\lambda_j^2}. \tag{121}
\end{aligned}
$$

In the thermodynamic limit this becomes

$$
\begin{aligned}
\frac{1}{L^3} \sum_{\lambda_a \in \Lambda} \sum_{\lambda_j \in \Lambda} \sum_{n \neq 0} \frac{f(\lambda_a, \lambda_j, \lambda_j + \frac{2\pi n}{L})}{(\lambda_j + \frac{2\pi n}{L})(-\frac{2\pi n}{L})} &= \int_{-\infty}^{\infty} \mathrm{d}\lambda \rho(\lambda) \fint \mathrm{d}\mu \frac{\rho(\mu)}{\mu} \fint \mathrm{d}\nu \frac{f(\lambda, \mu, \nu)}{2\pi\nu} \\
&- \int_{-\infty}^{\infty} \mathrm{d}\lambda \rho(\lambda) \fint \mathrm{d}\mu \frac{\rho(\mu)}{\mu} \fint \mathrm{d}\nu \frac{f(\lambda, \mu, \nu)}{2\pi(\nu - \mu)} - \Omega(\lambda) \int_{-\infty}^{\infty} f(\lambda, 0, 0)\rho(\lambda)\mathrm{d}\lambda. \tag{122}
\end{aligned}
$$

The two principal values can be brought under a single principal value as in (C.3), *cf.* Appendix A.2. Finally the fourth term in (119) can be calculated using Lemma 1 (101). Putting

everything together we then obtain

$$
\begin{aligned}
\Sigma_1'(x',t) &= \int_{-\infty}^{\infty} d\lambda \rho(\lambda) \fint d\mu \rho_h(\mu) \fint d\nu \rho_h(\nu) \frac{\nu-\lambda}{\lambda+\mu} e^{2it(\lambda^2-\mu^2)+ix'(\nu-\lambda)} \\
&+ \int_{-\infty}^{\infty} d\lambda \rho(\lambda) \fint d\nu \rho_h(\nu) \fint d\mu \rho_h(\mu) \frac{\lambda(\nu-\lambda)}{\nu(\mu-\nu)} e^{2it(\lambda^2-\mu^2)+ix'(\nu-\lambda)} \\
&+ \frac{1}{2}\left(\pi\rho(0) - \pi^2\rho(0)^2 - \tfrac{1}{4}\right) \int_{-\infty}^{\infty} \lambda^2 \rho(\lambda) e^{2it\lambda^2-ix'\lambda} d\lambda.
\end{aligned}
\tag{123}
$$

### 5.4.2  Second term $\Sigma_2'(x',t)$

The contribution to $\Sigma_2'(x',t)$ involving two simple poles is of the form

$$
\begin{aligned}
&\frac{1}{L^3} \sum_{\lambda_a \in \Lambda} \sum_{\mu_a \notin \Lambda} \sum_{\substack{\nu_a \notin \Lambda \\ \nu_a \neq \pm\mu_a}} \frac{f(\lambda_a, \mu_a, \nu_a)}{\mu_a(\lambda_a - \nu_a)} = \frac{1}{L^3} \sum_{\lambda_a \in \Lambda} \sum_m \sum_{n \neq 0} \frac{f\left(\lambda_a, \frac{2\pi(m+1/2)}{L}, \lambda_a + \frac{2\pi n}{L}\right)}{\left(-\frac{2\pi(m+1/2)}{L}\right)\frac{2\pi n}{L}} \\
&\quad - \frac{1}{L^3} \sum_{\lambda_a \in \Lambda} \sum_{\lambda_i \in \Lambda} \sum_{n \neq 0} \frac{f\left(\lambda_a, \lambda_i, \lambda_a + \frac{2\pi n}{L}\right)}{(-\lambda_i)\frac{2\pi n}{L}} - \frac{1}{L^3} \sum_{\lambda_a \in \Lambda} \sum_m \sum_{\substack{\lambda_j \in \Lambda \\ \lambda_j \neq \lambda_a}} \frac{f\left(\lambda_a, \frac{2\pi(m+1/2)}{L}, \lambda_j\right)}{\frac{2\pi(m+1/2)}{L}(\lambda_a - \lambda_j)} \\
&\quad + \frac{1}{L^3} \sum_{\lambda_a \in \Lambda} \sum_{\lambda_i \in \Lambda} \sum_{\substack{\lambda_j \in \Lambda \\ \lambda_j \neq \lambda_a}} \frac{f(\lambda_a, \lambda_i, \lambda_j)}{\lambda_i(\lambda_a - \lambda_j)} - \frac{1}{L^3} \sum_{\lambda_a \in \Lambda} \sum_{\mu_a \notin \Lambda} \left[ \frac{f(\lambda_a, \mu_a, \mu_a)}{\mu_a(\lambda_a - \mu_a)} + \frac{f(\lambda_a, \mu_a, -\mu_a)}{\mu_a(\lambda_a + \mu_a)} \right],
\end{aligned}
\tag{124}
$$

where

$$
f(\lambda, \mu, \nu) = \nu(\nu-\mu) e^{2it(\lambda^2-\mu^2)+ix'(\nu-\mu)}.
\tag{125}
$$

The first terms on the right-hand side can all be computed by performing successive one-dimensional sums with only a single simple pole, which allows them to be turned into principal value integrals in the thermodynamic limit. The last term involves a two-dimensional sum with a factor $L^{-3}$ and a summand featuring only simple poles. Hence it vanishes in the thermodynamic limit. We conclude that

$$
\begin{aligned}
\Sigma_2'(x',t) &= \int_{-\infty}^{\infty} d\lambda \rho(\lambda) \int_{-\infty}^{\infty} d\mu \rho_h(\mu) \fint d\nu \rho_h(\nu) \frac{\nu-\mu}{\mu+\lambda} e^{2it(\lambda^2-\mu^2)+ix'(\nu-\mu)} \\
&+ \int_{-\infty}^{\infty} d\lambda \rho(\lambda) \fint d\mu \rho_h(\mu) \fint d\nu \rho_h(\nu) \frac{\nu(\nu-\mu)}{\mu(\lambda-\nu)} e^{2it(\lambda^2-\mu^2)+ix'(\nu-\mu)} + \mathcal{O}(L^{-1}).
\end{aligned}
\tag{126}
$$

### 5.4.3  Third term $\Sigma_3'(x',t)$

This contribution is straightforward to deal with. After writing the sum over $\mu_a$ as the difference of a sum over vacancies and holes the sums over $\lambda_{a,b}$ can be factorized and will involve only single simple poles. It then follows that

$$
\begin{aligned}
\Sigma_3'(x',t) &= \int_{-\infty}^{\infty} d\lambda \rho(\lambda) \int_{-\infty}^{\infty} d\mu \rho_h(\mu) \fint d\nu \rho(\nu) \frac{\mu-\nu}{\lambda-\mu} e^{2it(\lambda^2-\mu^2)+ix'(\mu-\nu)} \\
&+ \int_{-\infty}^{\infty} d\lambda \rho(\lambda) \fint d\mu \rho_h(\mu) \fint d\nu \rho(\nu) \frac{\nu(\nu-\mu)}{\mu(\lambda-\nu)} e^{2it(\lambda^2-\mu^2)+ix'(\mu-\nu)} + \mathcal{O}(L^{-1}).
\end{aligned}
\tag{127}
$$

### 5.4.4 Fourth term $\Sigma'_4(x', t)$

The contribution to $\Sigma'_4(x', t)$ involving two simple poles is of the form

$$\frac{1}{L^3}\sum_{\lambda_a\in\Lambda}\sum_{\mu_a\notin\Lambda}\sum_{\lambda_b\in\Lambda}\frac{f(\lambda_a,\mu_a,\lambda_b)}{\lambda_b(\mu_a-\lambda_b)} = \frac{1}{L^3}\sum_{\lambda_a\in\Lambda}\sum_{n\neq0}\sum_{\lambda_b\in\Lambda}\frac{f(\lambda_a,\lambda_b+\frac{2\pi n}{L},\lambda_b)}{\lambda_b\frac{2\pi n}{L}}$$
$$-\frac{1}{L^3}\sum_{\lambda_a\in\Lambda}\sum_{\lambda_b\in\Lambda}\sum_{\substack{\lambda_i\in\Lambda\\\lambda_i\neq\lambda_b}}\frac{f(\lambda_a,\lambda_i,\lambda_b)}{\lambda_b(\lambda_i-\lambda_b)}, \tag{128}$$

where

$$f(\lambda,\mu,\nu) = \lambda(\nu-\lambda)e^{2it(\lambda^2-\mu^2)+ix'(\lambda-\nu)}. \tag{129}$$

The first sum can be straightforwardly turned into a principal value integral and the second sum can be carried out using Lemma 1 (101). This gives

$$\Sigma'_4(x', t) = \int_{-\infty}^{\infty}d\lambda\rho(\lambda)\int_{-\infty}^{\infty}d\mu\rho_h(\mu)\!\!\!\!\!\diagup\!\!\!\!\! d\nu\rho(\nu)\frac{\lambda-\nu}{\mu-\lambda}e^{2it(\lambda^2-\mu^2)+ix'(\lambda-\nu)}$$
$$+\int_{-\infty}^{\infty}d\lambda\rho(\lambda)\!\!\!\!\!\diagup\!\!\!\!\! d\mu\rho_h(\mu)\!\!\!\!\!\diagup\!\!\!\!\! d\nu\rho(\nu)\frac{\lambda(\nu-\lambda)}{\nu(\mu-\nu)}e^{2it(\lambda^2-\mu^2)+ix'(\lambda-\nu)}$$
$$+\frac{\pi^2\rho(0)^2-\Omega(\lambda)}{2}\int_{-\infty}^{\infty}\lambda^2\rho(\lambda)e^{2it\lambda^2+ix'\lambda}d\lambda. \tag{130}$$

### 5.4.5 Result for all contributions arising from type II configurations

The combined contribution of all $\Sigma'_n(x', t)$ can be brought into a simpler form by using that (i) the root distribution is even; (ii) at leading order in $1/c$ we can write

$$\rho(\lambda)+\rho_h(\lambda) = \frac{1}{2\pi}+\mathcal{O}(c^{-1}), \tag{131}$$

and (iii) for $x\neq0$ we have in a distribution sense

$$\int_{-\infty}^{\infty}e^{ix\nu}d\nu = 0, \qquad \int_{-\infty}^{\infty}\nu e^{ix\nu}d\nu = 0, \qquad \int_{-\infty}^{\infty}\frac{e^{ix\nu}}{\nu}d\nu = i\pi\,\text{sgn}(x). \tag{132}$$

This allows us to combine the contributions of the terms in $\Sigma'_n(x', t)$ involving only a single simple pole into the following expression

$$2\int_{-\infty}^{\infty}d\lambda\rho(\lambda)\!\!\!\!\!\diagup\!\!\!\!\! d\mu\rho_h(\mu)\!\!\!\!\!\diagup\!\!\!\!\! d\nu\rho(\nu)\frac{\nu-\lambda}{\lambda+\mu}\cos(x'(\nu-\lambda))e^{2it(\lambda^2-\mu^2)}$$
$$+2\int_{-\infty}^{\infty}d\lambda\rho(\lambda)\!\!\!\!\!\diagup\!\!\!\!\! d\mu\rho_h(\mu)\!\!\!\!\!\diagup\!\!\!\!\! d\nu\rho(\nu)\frac{\nu-\mu}{\lambda+\mu}\cos(x'(\nu-\mu))e^{2it(\lambda^2-\mu^2)}$$
$$-\text{sgn}(x)\int_{-\infty}^{\infty}d\lambda\rho(\lambda)\!\!\!\!\!\diagup\!\!\!\!\! d\mu\rho_h(\mu)\frac{\lambda(\lambda-\mu)}{\mu}\sin(x'(\lambda-\mu))e^{2it(\lambda^2-\mu^2)}. \tag{133}$$

Our final result for the thermodynamic limit of all contributions to (85) arising from type II

configurations is then

$$
\begin{aligned}
C_{\hat{\lambda}}^{2,1}(x',t) = &\frac{4}{c}\int_{-\infty}^{\infty}\mathrm{d}\lambda\rho(\lambda)\!\!\!\!\!\fint \mathrm{d}v\rho(v)\!\!\!\!\!\fint \mathrm{d}\mu\rho_h(\mu)\,F_2(\lambda,\mu,v;x')\cos\big(2t(\lambda^2-\mu^2)\big)\\
&-2\frac{\mathrm{sgn}(x')}{c}\int_{-\infty}^{\infty}\mathrm{d}\lambda\rho(\lambda)\!\!\!\!\!\fint \mathrm{d}\mu\rho_h(\mu)\frac{\lambda(\lambda-\mu)}{\mu}\sin(x(\lambda-\mu))e^{2it(\lambda^2-\mu^2)}\\
&+\frac{1}{c}\left(\frac{1}{4}-\pi\rho(0)+2\pi^2\rho(0)^2-\Omega(\lambda)\right)\int_{-\infty}^{\infty}\lambda^2\rho(\lambda)e^{2it\lambda^2-ix'\lambda}\mathrm{d}\lambda\,,\quad (134)
\end{aligned}
$$

where $F_2(\lambda,\mu,v;x')$ is the function defined in (24).

We stress that $C_{\hat{\lambda}}^{2,1}(x,t)$ depends on the representative state $\lambda$ not only through the root density $\rho$, but via the quantity $\Omega(\lambda)$ (102) as well.

## 5.5 Cancellation of the representative state dependence

Once both contributions (112) and (134) to the spectral sum are summed up, we observe that the dependence on the representative state through the quantity $\Omega(\lambda)$ exactly vanishes! This non-trivial cancellation suggests that the typicality assumption underlying (37) is indeed correct, even though the partial contributions do carry an additional dependence on the chosen representative state.

To arrive at the expression (22) written in the introduction, we sum up (112) and (134) and use that at leading order in $1/c$

$$
\rho(0) = \frac{1}{2\pi} + \mathcal{O}(c^{-1})\,. \tag{135}
$$

# 6 Calculation of the two-point function $\langle\sigma_2(x,\tau)\sigma_2(0,0)\rangle_\infty$ in the steady state

We saw in (50) that the expectation value of an observable $\langle\mathcal{O}\rangle_t$ after the quench converges when $t\to\infty$ to $\langle\mathcal{O}\rangle_\infty$ given in (49). This limit value is thus expressed as an equilibrium expectation value of $\mathcal{O}$ in a representative state corresponding to the steady-state root density $\rho$ that is fixed by the quench protocol. An interesting question is then how to characterize the physical properties of this steady state through its response functions.

The dynamical correlation function of an observable $\mathcal{O}$ in an energy eigenstate $|\lambda\rangle$ has a spectral representation in a basis of (unnormalized) energy eigenstates $|\mu\rangle$ of the form

$$
\langle\mathcal{O}(x,\tau)\mathcal{O}(0,0)\rangle = \sum_{\mu}\frac{|\langle\lambda|\mathcal{O}(0)|\mu\rangle|^2}{\langle\lambda|\lambda\rangle\langle\mu|\mu\rangle}e^{i\tau(E(\lambda)-E(\mu))+ix(P(\mu)-P(\lambda))}\,. \tag{136}
$$

We have previously considered the case where $\mathcal{O}(x)=\sigma(x)$ in [64]. This case is quite special as $\sigma(x)$ is the density of a conserved charge. In the following we consider the case $\mathcal{O}(x)=\sigma_2(x)$. An expression for the form factors of this operator between two states of equal momenta was presented previously in (47). To determine the dynamical two-point function we require form factors between states with different momenta as well, which can be expressed in the form [79]

$$
\frac{\langle\mu|\sigma_2(0)|\lambda\rangle}{\sqrt{\langle\lambda|\lambda\rangle\langle\mu|\mu\rangle}} = \frac{i}{6c}\frac{J(\lambda,\mu)}{(P(\lambda)-P(\mu))^2}\frac{\langle\mu|\sigma(0)|\lambda\rangle}{\sqrt{\langle\lambda|\lambda\rangle\langle\mu|\mu\rangle}}\,, \tag{137}
$$

where the density form factor given in (87) and

$$
J(\lambda,\mu) = (P(\lambda)-P(\mu))^4 - 4(P(\lambda)-P(\mu))(Q_3(\lambda)-Q_3(\mu)) + 3(E(\lambda)-E(\mu))^2\,. \tag{138}
$$

## 6.1 $1/c$ expansion and particle-hole excitations

Let us again follow the same reasoning as in the previous sections, and investigate the leading behaviour of the form factor when $c \to \infty$, for generic $\boldsymbol{\lambda}, \boldsymbol{\mu}$ satisfying the Bethe equations. The simple relation (137) allows us to directly use the results of [64] for the density correlations. Denoting $\nu$ the number of Bethe numbers of $\boldsymbol{\mu}$ that do not appear among those of $\boldsymbol{\lambda}$, we have

$$\left| \frac{\langle \boldsymbol{\mu} | \sigma_2(0) | \boldsymbol{\lambda} \rangle}{\sqrt{\langle \boldsymbol{\lambda} | \boldsymbol{\lambda} \rangle \langle \boldsymbol{\mu} | \boldsymbol{\mu} \rangle}} \right|^2 = \mathcal{O}(c^{-2\nu}). \tag{139}$$

Hence the $1/c$ expansion is also an expansion in the number of particle-hole excitations. By restricting our analysis to $\mathcal{O}(c^{-4})$, we can focus only on one and two-particle-hole excitations.

## 6.2 One particle-hole excitations

We now consider a one-particle-hole excitation above $\boldsymbol{\lambda}$, namely a state $\boldsymbol{\mu}$ such that all its Bethe numbers are those of $\boldsymbol{\lambda}$ except for $I_a$ which is replaced by $I_a + n$. This results in constraints on the Bethe numbers

$$n \neq 0, \qquad \forall i \neq a, I_a + n \neq I_i. \tag{140}$$

We then can use the results of [64] because of the simple relation (137), which always holds since the momenta between the two states involved are necessarily different. We obtain

$$\frac{|\langle \boldsymbol{\lambda} | \sigma_2(0) | \boldsymbol{\mu} \rangle|^2}{\langle \boldsymbol{\lambda} | \boldsymbol{\lambda} \rangle \langle \boldsymbol{\mu} | \boldsymbol{\mu} \rangle} = \frac{16}{c^4 L^2} \left[ -\mathcal{E} - \mathcal{D}\lambda_a^2 + 2\lambda_a \mathcal{P} + \frac{2\pi n}{L}(\mathcal{P} - \mathcal{D}\lambda_a) \right]^2 + \mathcal{O}(c^{-5}). \tag{141}$$

Interestingly, the a priori leading order $\mathcal{O}(c^{-2})$ contribution vanishes. As a result the one and two particle-hole excitations contribute at the same order in $1/c$. Since (141) does not have poles the corresponding contribution to the spectral sum (136) is straightforward to compute and gives the first line of (27).

## 6.3 Two particle-hole excitations

The other class of intermediate states contributing at order $\mathcal{O}(c^{-4})$ are two particle-hole excitations, i.e. states with rapidities $\boldsymbol{\mu}$ such that the corresponding Bethe numbers are those of $\boldsymbol{\lambda}$ with the exception of $I_a$ and $I_b$ which are replaced by $I_a + n$ and $I_b$ respectively. The Bethe numbers are subject to the following constraints

$$
\begin{aligned}
n, m &\neq 0, \\
\forall i \neq a, b, \quad I_a + n &\neq I_i, \quad I_b + m \neq I_i, \\
I_a + n &\neq I_b + m, \\
I_a + n &\neq I_b, \\
I_b + m &\neq I_a.
\end{aligned}
\tag{142}
$$

Assuming that the momenta of the two states are different, i.e. that $n \neq -m$, one can again use (137) and [64] to obtain

$$\frac{|\langle \boldsymbol{\lambda} | \sigma^2(0) | \boldsymbol{\mu} \rangle|^2}{\langle \boldsymbol{\lambda} | \boldsymbol{\lambda} \rangle \langle \boldsymbol{\mu} | \boldsymbol{\mu} \rangle} = \frac{16}{c^4 L^4} (\lambda_a - \lambda_b)^2 (\lambda_a - \lambda_b + \frac{2\pi(n-m)}{L})^2 + \mathcal{O}(c^{-5}). \tag{143}$$

This expression has no singularities and the corresponding contribution to the spectral sum is straightforwardly expressed as an integral in the thermodynamic limit. This gives the second line of (27).

When $n = -m$, i.e. when the momenta of the two states are identical, we obtain from (79) that

$$\frac{|\langle \boldsymbol{\lambda}|\sigma^2(0)|\boldsymbol{\mu}\rangle|^2}{\langle \boldsymbol{\lambda}|\boldsymbol{\lambda}\rangle\langle \boldsymbol{\mu}|\boldsymbol{\mu}\rangle} = \mathcal{O}(L^{-4}), \tag{144}$$

and that there are no singularities in $n$. As in this case there are only three sums we conclude that such contributions vanish in the thermodynamic limit.

# 7 Summary and Conclusions

In this work we have combined the Quench Action approach with our recently developed $1/c$-expansion method for form factor sums in the Lieb-Liniger model to analyze a number of different observables after a quantum quench starting in the ground state of a non-interacting Bose gas. To the best of our knowledge our work is the first to obtain analytic results for quench dynamics in a generic interacting integrable theory beyond the asymptotic late-time regime. This program has been carried out before only for the $q$-boson model in the limit $q \to \infty$ [80, 81], an interacting model that exhibits unusual simplifying features, and some other particular situations [82–84].

Our work also uncovered a novel aspect regarding the application of typicality ideas to the analysis of quantum quenches in integrable models. We observed that carrying out partial summations of the spectral sums in the Quench Action approach can lead to results that violate the underlying typicality assumption and depend on details of the particular representative state selected. In the case at hand this dependence arises from the singular behaviour of overlaps at zero rapidity. But remarkably, we observe that this representative-state dependence cancels out between different types of particle-hole excitations at the order in $1/c$ of our calculation, yielding a significant check of typicality in an out-of-equilibrium setting. However, we are able to construct *ad hoc* initial states in a free theory for which these cancellations do not occur. This results in a failure of typicality, but this failure is weak in the sense that the problematic representative states are rare and can be avoided through a regularization procedure. A brief discussion of these findings is given in Appendix E.

Our work raises a number of interesting questions that should be investigated further. First, it is important to work out higher orders in the $1/c$-expansion both for dynamical response functions and in the quench context. In particular, conjectured extensions of GHD predict that the two-point functions of $\sigma_2(x)$ will exhibit diffusive behaviour [71]. This is not seen in the leading order of the $1/c$-expansion worked out here, but supposedly will appear at the next order. Second, it should be explored how to define truncations of the spectral sum that would be finite in the thermodynamic limit (not divergent and not exponentially small) for finite $c$. Indeed, the spectral sum truncation induced by the $1/c$ expansion generically exhibits terms polynomial in the system size that cross-cancel between different numbers of particle-hole excitations. Third, it would be very interesting to apply our strong coupling expansion method to dynamical correlations in other models like the Heisenberg XXZ chain [85–88]. These typically will involve bound states, and an important question is how to extend the strong coupling expansion in order to take their contributions into account. Fourth, it would be interesting to extend the analysis presented above to quantum quenches starting in inhomogeneous initial states [89, 90]. Finally, we think it is important to arrive at a more complete understanding of the scope and limitations for applying typicality ideas to the calculation of dynamical correlations in and out of equilibrium.

**Acknowledgements** We are grateful to Jacopo de Nardis and Karol Kozlowski for helpful discussions and comments. This work was supported by the EPSRC under grant EP/S020527/1.

# A Principal value integrals

In this appendix we present details on principal value integrals used in the main text and the proofs of Lemma 1 and 2.

## A.1 Double principal values

Given a function $F(\lambda, \mu, \nu)$, we define its integral with successive double principal value as

$$\fint \frac{F(\lambda,\mu,\nu)}{(\lambda-\mu)(\mu-\nu)}\mathrm{d}\lambda\mathrm{d}\mu\mathrm{d}\nu = \int \mathrm{d}\mu \fint \mathrm{d}\nu \frac{1}{\mu-\nu}\fint \mathrm{d}\lambda \frac{F(\lambda,\mu,\nu)}{\lambda-\mu}, \tag{A.1}$$

where the $\fint$ symbols appearing in the right-hand side of this expression denote single principal values defined in (25). As shown in [64], the following relations hold

$$\fint \frac{F(\lambda,\mu,\nu)}{(\lambda-\mu)(\mu-\nu)}\mathrm{d}\lambda\mathrm{d}\mu\mathrm{d}\nu = \int \mathrm{d}\nu \fint \mathrm{d}\mu \frac{1}{\mu-\nu}\fint \mathrm{d}\lambda \frac{F(\lambda,\mu,\nu)}{\lambda-\mu}$$

$$= \int \mathrm{d}\lambda \fint \mathrm{d}\mu \frac{1}{\lambda-\mu}\fint \mathrm{d}\nu \frac{F(\lambda,\mu,\nu)}{\mu-\nu} \tag{A.2}$$

$$= \int \mathrm{d}\mu \fint \mathrm{d}\lambda \frac{1}{\lambda-\mu}\fint \mathrm{d}\nu \frac{F(\lambda,\mu,\nu)}{\mu-\nu},$$

and

$$\fint \frac{F(\lambda,\mu,\nu)}{(\lambda-\mu)(\mu-\nu)}\mathrm{d}\lambda\mathrm{d}\mu\mathrm{d}\nu = \lim_{\epsilon,\epsilon'\to 0} \int_{\substack{|\lambda-\mu|>\epsilon \\ |\mu-\nu|>\epsilon'}} \frac{F(\lambda,\mu,\nu)}{(\lambda-\mu)(\mu-\nu)}\mathrm{d}\lambda\mathrm{d}\mu\mathrm{d}\nu. \tag{A.3}$$

The integral with simultaneous double principal value is defined by

$$\oiint \frac{F(\lambda,\mu,\nu)}{(\lambda-\mu)(\mu-\nu)}\mathrm{d}\lambda\mathrm{d}\mu\mathrm{d}\nu = \lim_{\epsilon\to 0} \int_{\substack{|\lambda-\mu|>\epsilon \\ |\mu-\nu|>\epsilon \\ |\lambda-\nu|>\epsilon}} \frac{F(\lambda,\mu,\nu)}{(\lambda-\mu)(\mu-\nu)}\mathrm{d}\lambda\mathrm{d}\mu\mathrm{d}\nu. \tag{A.4}$$

As shown in [64], it is related to the integral with successive double principal value through the Poincaré-Bertrand-like formula

$$\oiint \frac{F(\lambda,\mu,\nu)}{(\lambda-\mu)(\mu-\nu)}\mathrm{d}\lambda\mathrm{d}\mu\mathrm{d}\nu = \fint \frac{F(\lambda,\mu,\nu)}{(\lambda-\mu)(\mu-\nu)}\mathrm{d}\lambda\mathrm{d}\mu\mathrm{d}\nu + \frac{\pi^2}{3}\int_{-\infty}^{\infty} F(\lambda,\lambda,\lambda)\mathrm{d}\lambda. \tag{A.5}$$

## A.2 Proof of equation (C.3)

Using the identity (A.5) we obtain

$$\int_{-\infty}^{\infty}\mathrm{d}\lambda \fint \mathrm{d}\mu \frac{1}{\mu}\fint \mathrm{d}\nu \frac{F(\lambda,\mu,\nu)}{\nu} = -\fint \frac{F(\lambda,\mu-\lambda,\nu-\lambda)}{(\nu-\lambda)(\lambda-\mu)}\mathrm{d}\lambda\mathrm{d}\mu\mathrm{d}\nu$$

$$= -\oiint \frac{F(\lambda,\mu-\lambda,\nu-\lambda)}{(\nu-\lambda)(\lambda-\mu)}\mathrm{d}\lambda\mathrm{d}\mu\mathrm{d}\nu + \frac{\pi^2}{3}\int_{-\infty}^{\infty} F(\lambda,0,0)\mathrm{d}\lambda, \tag{A.6}$$

and

$$\int_{-\infty}^{\infty}\mathrm{d}\lambda \fint \mathrm{d}\mu \frac{1}{\mu}\fint \mathrm{d}\nu \frac{F(\lambda,\mu,\nu)}{\nu-\mu} = \fint \frac{F(\lambda,\mu-\lambda,\nu-\lambda)}{(\nu-\mu)(\mu-\lambda)}\mathrm{d}\lambda\mathrm{d}\mu\mathrm{d}\nu$$

$$= \oiint \frac{F(\lambda,\mu-\lambda,\nu-\lambda)}{(\nu-\mu)(\mu-\lambda)}\mathrm{d}\lambda\mathrm{d}\mu\mathrm{d}\nu - \frac{\pi^2}{3}\int_{-\infty}^{\infty} F(\lambda,0,0)\mathrm{d}\lambda. \tag{A.7}$$

In (C.3) the sum of these two quantities appears. The latter can be brought under a single simultaneous principal value because the excluded regions of the integral are identical (which is not the case of the successive principal values). Hence

$$
\int_{-\infty}^{\infty} d\lambda \fint d\mu \frac{1}{\mu} \fint d\nu \frac{F(\lambda,\mu,\nu)}{\nu} - \int_{-\infty}^{\infty} d\lambda \fint d\mu \frac{1}{\mu} \fint d\nu \frac{F(\lambda,\mu,\nu)}{\nu-\mu}
$$
$$
= \fint \frac{F(\lambda,\mu-\lambda,\nu-\lambda)}{(\lambda-\nu)(\nu-\mu)} d\lambda d\mu d\nu + \frac{2\pi^2}{3} \int_{-\infty}^{\infty} F(\lambda,0,0) d\lambda
$$
$$
= \fint \frac{F(\lambda,\mu-\lambda,\nu-\lambda)}{(\lambda-\nu)(\nu-\mu)} d\lambda d\mu d\nu + \pi^2 \int_{-\infty}^{\infty} F(\lambda,0,0) d\lambda \, .
$$

(A.8)

Using (A.2) we arrive at (C.3).

## B  Proof of Lemma 1 (101)

We start by adding the condition $\lambda_j \neq -\lambda_i$

$$
\frac{1}{L^3} \sum_{\lambda_a \in \Lambda} \sum_{\lambda_i \in \Lambda} \sum_{\substack{\lambda_j \in \Lambda \\ \lambda_j \neq \lambda_i}} \frac{f(\lambda_a,\lambda_j,\lambda_i)}{\lambda_i(\lambda_j-\lambda_i)} = \frac{1}{L^3} \sum_{\lambda_a \in \Lambda} \sum_{\lambda_i \in \Lambda} \sum_{\substack{\lambda_j \in \Lambda \\ \lambda_j \neq \lambda_i,-\lambda_i}} \frac{f(\lambda_a,\lambda_j,\lambda_i)}{\lambda_i(\lambda_j-\lambda_i)}
$$
$$
- \frac{1}{2L^3} \sum_{\lambda_a \in \Lambda} \sum_{\lambda_i \in \Lambda} \frac{f(\lambda_a,-\lambda_i,\lambda_i)}{\lambda_i^2} \, .
$$

(B.1)

The second sum is two-dimensional and comes with a factor $L^{-3}$. Hence it is dominated by the double pole and its thermodynamic limit reads

$$
\frac{1}{2L^3} \sum_{\lambda_a \in \Lambda} \sum_{\lambda_i \in \Lambda} \frac{f(\lambda_a,-\lambda_i,\lambda_i)}{\lambda_i^2} = \frac{\Omega(\boldsymbol{\lambda})}{2} \int_{-\infty}^{\infty} f(\lambda,0,0)\rho(\lambda)d\lambda + \mathcal{O}(L^{-1}) \, .
$$

(B.2)

To compute the first term on the right-hand side in (B.1) we symmetrize in $i, j$ and $\pm\lambda_{i,j}$ using the pair structure of the state

$$
\frac{1}{L^3} \sum_{\lambda_a \in \Lambda} \sum_{\lambda_i \in \Lambda} \sum_{\substack{\lambda_j \in \Lambda \\ \lambda_j \neq \lambda_i,-\lambda_i}} \frac{f(\lambda_a,\lambda_j,\lambda_i)}{\lambda_i(\lambda_j-\lambda_i)} = \frac{1}{8L^3} \sum_{\lambda_a \in \Lambda} \sum_{\lambda_i \in \Lambda} \sum_{\substack{\lambda_j \in \Lambda \\ \lambda_j \neq \lambda_i,-\lambda_i}} G(\lambda_a,\lambda_j,\lambda_i) \, .
$$

(B.3)

Here we have defined

$$
G(\lambda_a,\lambda_j,\lambda_i) = \frac{g(\lambda_a,\lambda_j,\lambda_i) - g(\lambda_a,-\lambda_j,\lambda_i) - g(\lambda_a,\lambda_j,-\lambda_i) + g(\lambda_a,-\lambda_j,-\lambda_i)}{\lambda_i \lambda_j} \, ,
$$
$$
g(\lambda_a,\lambda_j,\lambda_i) = \frac{\lambda_j f(\lambda_a,\lambda_j,\lambda_i) - \lambda_i f(\lambda_a,\lambda_i,\lambda_j)}{\lambda_j - \lambda_i} \, .
$$

(B.4)

The right-hand side in (B.3) is a Riemann sum of a regular function without singularities, hence converges to an integral in the thermodynamic limit

$$
\frac{1}{L^3} \sum_{\lambda_a \in \Lambda} \sum_{\lambda_i \in \Lambda} \sum_{\substack{\lambda_j \in \Lambda \\ \lambda_j \neq \pm\lambda_i}} \frac{f(\lambda_a,\lambda_j,\lambda_i)}{\lambda_i(\lambda_j-\lambda_i)} = \frac{1}{8} \iiint_{-\infty}^{\infty} G(x,y,z)\,\rho(x)\rho(y)\rho(z) dx dy dz + \mathcal{O}(L^{-1}) \, .
$$

(B.5)

To proceed, we remove from the integration region the points where $|y| < \epsilon$ or $|z| < \epsilon$. This incurs only an error $\mathcal{O}(\epsilon)$ since the integrand is regular and allows us to split the integral into four. We then replace $y$ and $z$ by $y - x$ and $z - x$ and use (A.3) to obtain

$$\frac{1}{L^3} \sum_{\lambda_a \in \Lambda} \sum_{\lambda_i \in \Lambda} \sum_{\substack{\lambda_j \in \Lambda \\ \lambda_j \neq \pm \lambda_i}} \frac{f(\lambda_a, \lambda_j, \lambda_i)}{\lambda_i (\lambda_j - \lambda_i)} = -\frac{1}{2} \int \frac{g(x, y-x, z-x) \rho(x) \rho(y-x) \rho(z-x)}{(y-x)(x-z)} \mathrm{d}x \mathrm{d}y \mathrm{d}z .$$

(B.6)

Under the successive principal value we cannot use the definition of $g$ in terms of $f$ and split the integral into two since we do not necessarily have $|z - y| > \epsilon$. However, we can use (A.5) to obtain an expression in terms of a simultaneous principal value integral

$$\frac{1}{L^3} \sum_{\lambda_a \in \Lambda} \sum_{\lambda_i \in \Lambda} \sum_{\substack{\lambda_j \in \Lambda \\ \lambda_j \neq \pm \lambda_i}} \frac{f(\lambda_a, \lambda_j, \lambda_i)}{\lambda_i (\lambda_j - \lambda_i)} = -\frac{1}{2} \fint \frac{g(x, y-x, z-x) \rho(x) \rho(y-x) \rho(z-x)}{(y-x)(x-z)} \mathrm{d}x \mathrm{d}y \mathrm{d}z$$

$$+ \frac{\pi^2 \rho(0)^2}{6} \int_{-\infty}^{\infty} f(x, 0, 0) \rho(x) \mathrm{d}x .$$

(B.7)

We now express $g$ in terms of $f$, split the integral and swap the variables $y, z$ in one of the two resulting integrals to obtain

$$\frac{1}{L^3} \sum_{\lambda_a \in \Lambda} \sum_{\lambda_i \in \Lambda} \sum_{\substack{\lambda_j \in \Lambda \\ \lambda_j \neq \pm \lambda_i}} \frac{f(\lambda_a, \lambda_j, \lambda_i)}{\lambda_i (\lambda_j - \lambda_i)} = \fint \frac{f(x, y-x, z-x) \rho(x) \rho(y-x) \rho(z-x)}{(y-z)(z-x)} \mathrm{d}x \mathrm{d}y \mathrm{d}z$$

$$+ \frac{\pi^2 \rho(0)^2}{6} \int_{-\infty}^{\infty} f(x, 0, 0) \rho(x) \mathrm{d}x .$$

(B.8)

Finally we employ (A.5) to arrive at Eq (101).

## C Proof of Lemma 2 (103)

We start by rewriting the multiple sum of interest as

$$\frac{1}{L^3} \sum_{\lambda_a \in \Lambda} \sum_{n \neq 0} \sum_{\lambda_i \in \Lambda} \frac{f(\lambda_a, \lambda_i + \frac{2\pi n}{L}, \lambda_i)}{(\lambda_i + \frac{2\pi n}{L})(-\frac{2\pi n}{L})} = -\frac{1}{L^3} \sum_{\lambda_a \in \Lambda} \sum_{n \neq 0} \sum_{\lambda_i \in \Lambda} \frac{f(\lambda_a, \lambda_i + \frac{2\pi n}{L}, \lambda_i)}{\lambda_i \frac{2\pi n}{L}}$$

$$+ \frac{1}{L^3} \sum_{\lambda_a \in \Lambda} \sum_{n} \sum_{\lambda_i \in \Lambda} \frac{f(\lambda_a, \lambda_i + \frac{2\pi n}{L}, \lambda_i)}{(\lambda_i + \frac{2\pi n}{L}) \lambda_i} - \frac{1}{L^3} \sum_{\lambda_a \in \Lambda} \sum_{\lambda_i \in \Lambda} \frac{f(\lambda_a, \lambda_i, \lambda_i)}{\lambda_i^2} .$$

(C.1)

The first and second terms on the right-hand side can be turned into principal part integrals in the thermodynamic limit by first summing over $n$ and then over $\lambda_i$. The third sum, although two-dimensional with a prefactor $1/L^3$, is not negligible in the thermodynamic limit since it involves a double pole in $\lambda_i$. Its thermodynamic in fact depends on the representative state $\boldsymbol{\lambda}$ through the quantity $\Omega(\boldsymbol{\lambda})$ defined in (102).

$$\frac{1}{L^3} \sum_{\lambda_a \in \Lambda} \sum_{n \neq 0} \sum_{\lambda_i \in \Lambda} \frac{f(\lambda_a, \lambda_i + \frac{2\pi n}{L}, \lambda_i)}{(\lambda_i + \frac{2\pi n}{L})(-\frac{2\pi n}{L})} = -\frac{1}{2\pi} \int_{-\infty}^{\infty} \mathrm{d}\lambda \rho(\lambda) \fint \mathrm{d}\nu \frac{\rho(\nu)}{\nu} \fint \mathrm{d}\mu \frac{1}{\mu - \nu} f(\lambda, \mu, \nu)$$

$$+ \frac{1}{2\pi} \int_{-\infty}^{\infty} \mathrm{d}\lambda \rho(\lambda) \fint \mathrm{d}\nu \frac{\rho(\nu)}{\nu} \fint \mathrm{d}\mu \frac{1}{\mu} f(\lambda, \mu, \nu) - \Omega(\boldsymbol{\lambda}) \int_{-\infty}^{\infty} \rho(\lambda) f(\lambda, 0, 0) \mathrm{d}\lambda .$$

(C.2)

The two principal values can be brought under a single principal value according to the following relation, proved in Appendix A.2

$$\int_{-\infty}^{\infty} d\lambda \rho(\lambda) \fint d\mu \frac{\rho(\mu)}{\mu} \fint dv \frac{f(\lambda, \mu, v)}{v} - \int_{-\infty}^{\infty} d\lambda \rho(\lambda) \fint d\mu \frac{\rho(\mu)}{\mu} \fint dv \frac{f(\lambda, \mu, v)}{v - \mu}$$

$$= \int_{-\infty}^{\infty} d\lambda \rho(\lambda) \fint dv \fint d\mu \rho(\mu) \frac{f(\lambda, \mu, v)}{v(\mu - v)} + \pi^2 \rho(0) \int_{-\infty}^{\infty} f(\lambda, 0, 0) \rho(\lambda) d\lambda. \qquad (C.3)$$

This gives the desired result

$$\frac{1}{L^3} \sum_{\lambda_a \in \Lambda} \sum_{n \neq 0} \sum_{\lambda_i \in \Lambda} \frac{f(\lambda_a, \lambda_i + \frac{2\pi n}{L}, \lambda_i)}{(\lambda_i + \frac{2\pi n}{L})(-\frac{2\pi n}{L})} = \frac{1}{2\pi} \int_{-\infty}^{\infty} d\lambda \rho(\lambda) \fint dv \rho(v) \fint d\mu \frac{f(\lambda, \mu, v)}{\mu(v - \mu)}$$

$$+ \frac{1}{2\pi} \rho(0) \pi^2 \int_{-\infty}^{\infty} f(\lambda, 0, 0) \rho(\lambda) d\lambda - \Omega(\boldsymbol{\lambda}) \int_{-\infty}^{\infty} \rho(\lambda) f(\lambda, 0, 0) d\lambda. \qquad (C.4)$$

# D  Further results on $\langle \sigma_2(x) \sigma_2(0) \rangle$

In this appendix we collect a number of additional results on the two-point function after our interaction quench (22).

## D.1  Alternative expression for (22)

In this section we present an alternative expression for the two-point function after the quench (22), that is particularly useful for numerical purposes. It is based on the observation that the first terms in the $1/c$-expansion of the steady state root density (15) take the simple form

$$\rho_s(\lambda) = \frac{1 + \frac{2\mathcal{D}}{c}}{2\pi} \frac{1}{\left(\frac{\lambda}{2\mathcal{D}\left[1 + \frac{2\mathcal{D}}{c}\right]}\right)^2 + 1} + \mathcal{O}(c^{-2}), \qquad (D.1)$$

which allows one to carry out some of the integrals in (22). To that end we introduce

$$\mathcal{I}_{x,t}[f(\lambda)] = \int_{-\infty}^{\infty} e^{-ix\lambda + 2it\lambda^2} f(\lambda) d\lambda. \qquad (D.2)$$

We then find

$$\langle \sigma(x)\sigma(0) \rangle_t - \langle \sigma(x)\sigma(0) \rangle_\infty = (1 + \tfrac{2\mathcal{D}}{c})^6 \left(\frac{\mathcal{D}}{\pi}\right)^2 F_0(\bar{x}, \bar{t}) + \frac{4\pi}{c}(1 + \tfrac{2\mathcal{D}}{c})^6 \left(\frac{\mathcal{D}}{\pi}\right)^3 \mathrm{Re} F_1(\bar{x}, \bar{t}), \qquad (D.3)$$

where we have defined $\bar{x} = 2\mathcal{D}(1 + \tfrac{2\mathcal{D}}{c})^2 x$, $\bar{t} = [2\mathcal{D}(1 + \tfrac{2\mathcal{D}}{c})]^2 t$,

$$F_0(x, t) = \left| \mathcal{I}_{x,t}\left[\frac{\lambda}{1+\lambda^2}\right] \right|^2,$$

$$F_1(x, t) = 2\left( \mathcal{I}_{x,t}\left[\frac{\lambda^2}{1+\lambda^2}\right] \mathcal{I}_{x,t}\left[\frac{\lambda^2}{(1+\lambda^2)^2}\right]^* - \mathcal{I}_{x,t}\left[\frac{\lambda}{1+\lambda^2}\right] \mathcal{I}_{x,t}\left[\frac{\lambda^3}{(1+\lambda^2)^2}\right]^* \right)$$

$$+ i \, \mathrm{sgn}(x)\left( \mathcal{I}_{x,t}\left[\frac{\lambda}{1+\lambda^2}\right] \mathcal{I}_{x,t}\left[\frac{\lambda^2}{1+\lambda^2}\right]^* - \mathcal{I}_{x,t}\left[\frac{\lambda^2}{1+\lambda^2}\right] \mathcal{I}_{x,t}\left[\frac{\lambda}{1+\lambda^2}\right]^* \right)$$

$$+ 2i \, \mathrm{sgn}(x)\left( \mathcal{I}_{x,t}\left[\frac{\lambda^2}{1+\lambda^2}\right] \mathcal{I}_{x,t}\left[\frac{\lambda}{(1+\lambda^2)^2}\right]^* - \mathcal{I}_{x,t}\left[\frac{\lambda}{1+\lambda^2}\right] \mathcal{I}_{x,t}\left[\frac{\lambda^2}{(1+\lambda^2)^2}\right]^* \right)$$

$$+ 2e^{-|x|}\left( \mathcal{I}_{x,t}\left[\frac{\lambda^2}{1+\lambda^2}\right] \mathcal{I}_{0,t}\left[\frac{1}{(1+\lambda^2)^2}\right]^* - \mathcal{I}_{x,t}\left[\frac{\lambda^4}{(1+\lambda^2)^2}\right] \mathcal{I}_{0,t}\left[\frac{1}{1+\lambda^2}\right]^* \right)$$

$$+ 2i \, \mathrm{sgn}(x)e^{-|x|}\left( \mathcal{I}_{x,t}\left[\frac{\lambda^3}{(1+\lambda^2)^2}\right] \mathcal{I}_{0,t}\left[\frac{1}{1+\lambda^2}\right]^* - \mathcal{I}_{x,t}\left[\frac{\lambda}{1+\lambda^2}\right] \mathcal{I}_{0,t}\left[\frac{1}{(1+\lambda^2)^2}\right]^* \right)$$

$$- 2e^{-|x|} \mathcal{I}_{x,t}\left[\frac{\lambda^3}{(1+\lambda^2)^2} H_t(\lambda)\right] + 2i \, \mathrm{sgn}(x)e^{-|x|} \mathcal{I}_{x,t}\left[\frac{\lambda^2}{(1+\lambda^2)^2} H_t(\lambda)\right], \qquad (D.4)$$

and

$$H_t(\lambda) = \int \frac{e^{-2it\mu^2}}{\lambda - \mu} d\mu. \tag{D.5}$$

## D.2 Consistency check I: $t \to 0$ limit

Since the expression (22) for the two-point function $\langle \sigma(x)\sigma(0) \rangle_t$ holds for all $t > 0$ it should be possible to take the limit $t \to 0$ and recover the order $\mathcal{O}(c^{-2})$ result for the corresponding correlation function within the BEC state. The latter are simple

$$\langle \Psi_{\text{BEC}} | \sigma(x)\sigma(0) | \Psi_{\text{BEC}} \rangle = \mathcal{D}^2. \tag{D.6}$$

In order to investigate the $t \to 0$ limit of (22) we require an explicit expression at order $\mathcal{O}(c^{-2})$ for its infinite time limit $\langle \sigma(x)\sigma(0) \rangle_\infty$. Using [64] we find

$$\langle \sigma(x)\sigma(0) \rangle_\infty = \mathcal{D}^2 - \mathcal{D}^2 e^{-4\mathcal{D}(1+\frac{2\mathcal{D}}{c})^2|x|} \left( 1 + \frac{16\mathcal{D}^2}{c}|x| \right) + \mathcal{O}(c^{-2}). \tag{D.7}$$

At $t = 0$, all integrals appearing in (D.4) can be carried out explicitly by noting that

$$\int_{-\infty}^{\infty} \frac{e^{ix\lambda}}{(1+\lambda^2)^2} d\lambda = \frac{\pi}{2}(1+|x|)e^{-|x|}. \tag{D.8}$$

The integrals in (D.4) can be deduced by differentiating this with respect to $x$. A straightforward calculation then shows that at $t = 0$ we indeed recover the two-point function in the BEC state at order $\mathcal{O}(c^{-2})$

$$\langle \sigma(x)\sigma(0) \rangle_{t=0} = \mathcal{D}^2 + \mathcal{O}(c^{-2}). \tag{D.9}$$

## D.3 Consistency check II: $x \to 0$ limit

Since for $x \neq 0$ we have

$$\sigma(x)\sigma(0) = \psi^\dagger(x)\psi^\dagger(0)\psi(x)\psi(0), \tag{D.10}$$

the correlation function $\langle \sigma(x)\sigma(0) \rangle_t$ should approach $\langle \sigma_2(0) \rangle_t = \mathcal{O}(c^{-2})$ in the $x \to 0$ limit. Using (D.7) at $x = 0$ and simplifying (22) by exploiting that the root density $\rho(\lambda)$ is even we find after some calculations that indeed

$$\lim_{x \to 0} \langle \sigma(x)\sigma(0) \rangle_t = \mathcal{O}(c^{-2}). \tag{D.11}$$

## D.4 Some remarks on the limit $x, t \to 0$

As we have noted in the main text the limit $t \to 0$ of our result for $\langle \sigma_2(0) \rangle_t$ does not recover the correct result for the expectation value of $\sigma_2$ in the BEC initial state, $\mathcal{D}^2$. On the other hand, we have just shown that the limit $x \to 0$ of $\langle \sigma(x)\sigma(0) \rangle_{t=0}$ does reduce to $\mathcal{D}^2$. On a technical level it can be traced back to properties of the integral

$$\left| \int_{-\infty}^{\infty} \frac{\lambda}{1+\lambda^2} e^{-ix\lambda+2it\lambda^2} d\lambda \right|^2, \tag{D.12}$$

which vanishes if one first takes the limit $x \to 0$ and then $t \to 0$, but gives a finite result if one takes first $t \to 0$ and then $x \to 0$.

# E  Typicality and Quench Action method

In this Appendix we present an *ad hoc* initial state in a free theory for which the Quench Action spectral sum for the out-of-equilibrium dynamics is representative state dependent. We consider a simple tight-binding Hamiltonian on a ring

$$H = \sum_{j=1}^{L} a_j^\dagger a_{j+1} + a_{j+1}^\dagger a_j - 2a_j^\dagger a_j \,, \tag{E.1}$$

where $a_j^\dagger, a_j$ are fermionic creation and annihilation operators satisfying canonical anticommutation relations $\{a_j, a_k^\dagger\} = \delta_{j,k}$. The Hamiltonian is straightforwardly diagonalized by a canonical transformation to Bogoliubov fermions in momentum space

$$H = -4 \sum_{n=1}^{L} \sin^2(k_n/2) b_{k_n}^\dagger b_{k_n} \,, \tag{E.2}$$

where $k_n = \frac{2\pi n}{L}$ and $\{b_p, b_k^\dagger\} = \delta_{p,k}$. We denote the Bogoliubov vacuum state by $|0\rangle$. We now consider a quantum quench where the system is initialized in a Gaussian state parametrized by a fixed arbitrary function $K(p)$

$$|I\rangle = \prod_{m=1}^{L/2-1} \frac{1}{\sqrt{1 + K^2(k_m)}} \exp\left[ i \sum_{n=1}^{L/2-1} K(k_n) b_{-k_n}^\dagger b_{k_n}^\dagger \right] |0\rangle \,. \tag{E.3}$$

For our purposes it is sufficient to focus on the Green's function

$$G(n,t) = \langle I(t)| a_{n+1} a_1 |I(t)\rangle \,. \tag{E.4}$$

Since the model is free $G(n,t)$ can be straightforwardly calculated

$$G(n,t) = \frac{1}{2\pi} \int_{-\pi}^{\pi} \frac{iK(k)}{1 + K^2(k)} e^{8it \sin^2(k/2)} e^{ikn} \mathrm{d}k + \mathcal{O}(L^{-1}) \,. \tag{E.5}$$

Let us now try to recover this with the Quench Action approach. The normalized overlaps of the initial state with an eigenstate $|\bar{\boldsymbol{\lambda}}\rangle = \prod_{k \in \boldsymbol{\lambda}} b_{-k}^\dagger b_k^\dagger |0\rangle$ are

$$\langle \bar{\boldsymbol{\lambda}} | I \rangle = \frac{\prod_{k \in \boldsymbol{\lambda}} iK(k)}{\prod_{n=1}^{L/2-1} \sqrt{1 + K^2(k_n)}} \,, \tag{E.6}$$

from which one finds the root density characterizing the non-equilibrium steady state reached at late times after the quench

$$\rho(k) = \frac{1}{2\pi} \frac{K^2(k)}{1 + K^2(k)} \,. \tag{E.7}$$

The form factor of the operator of interest between two pair states $\bar{\boldsymbol{\lambda}}, \bar{\boldsymbol{\mu}}$ is

$$\langle \bar{\boldsymbol{\lambda}} | a_{n+1} a_1 | \bar{\boldsymbol{\mu}} \rangle = \begin{cases} \frac{e^{2i\pi kn}}{L} & \text{if } \boldsymbol{\mu} = \boldsymbol{\lambda} \cup \{k\} \text{ and } k \notin \boldsymbol{\lambda} \,, \\ 0 & \text{else} \,. \end{cases} \tag{E.8}$$

Let us now choose a representative pair state $\boldsymbol{\lambda}$ of the root density $\rho$, and write the Quench Action spectral sum

$$\begin{aligned} \frac{\langle \bar{\boldsymbol{\lambda}} | (a_{n+1} a_1)(t) | I \rangle}{\langle \bar{\boldsymbol{\lambda}} | I \rangle} &= \sum_{\boldsymbol{\mu}} \langle \bar{\boldsymbol{\lambda}} | a_{n+1} a_1 | \bar{\boldsymbol{\mu}} \rangle \frac{\langle \bar{\boldsymbol{\mu}} | I \rangle}{\langle \bar{\boldsymbol{\lambda}} | I \rangle} e^{2it(E(\boldsymbol{\lambda}) - E(\boldsymbol{\mu}))} \\ &= \frac{1}{L} \sum_{k \notin \boldsymbol{\lambda}} iK(k) e^{8it \sin^2(k/2)} e^{ikn} \,. \end{aligned} \tag{E.9}$$

If $K(k)$ is a regular function of $k$ this sum can be turned into an integral over the density of holes

$$\rho_h(k) = \frac{1}{2\pi} \frac{1}{1 + K^2(k)}, \tag{E.10}$$

and the Quench Action approach precisely recovers the result (E.5)

$$\lim_{L \to \infty} \frac{\langle \bar{\boldsymbol{\lambda}} | (a_{n+1} a_1)(t) | I \rangle}{\langle \bar{\boldsymbol{\lambda}} | I \rangle} = \frac{1}{2\pi} \int_{-\pi}^{\pi} \frac{iK(k)}{1 + K^2(k)} e^{8it \sin^2(k/2)} e^{ikn} \mathrm{d}k . \tag{E.11}$$

So far we have closely followed the discussion in [24]. However, let us now consider the following singular behaviour

$$K(k) = \frac{1}{k^m}, \tag{E.12}$$

with $m \geq 1$ an integer, and define a representative state $|\boldsymbol{\lambda}'\rangle$ by replacing $k_0 \in \boldsymbol{\lambda}$ by $k_0'$. By construction $|\boldsymbol{\lambda}'\rangle$ is a micro-state that for any choice of $k_0, k_0'$ corresponds to the macro-state with particle density $\rho$ in the thermodynamic limit, and in particular the extensive parts of all local conservation laws are the same for $|\boldsymbol{\lambda}'\rangle$ and $|\boldsymbol{\lambda}\rangle$. Let us choose $k_0'$ finite in the thermodynamic limit, and $k_0 = \mathcal{O}(L^{-1})$. We observe that

$$\frac{\langle \bar{\boldsymbol{\lambda}} | (a_{n+1} a_1)(t) | I \rangle}{\langle \bar{\boldsymbol{\lambda}} | I \rangle} = \frac{\langle \bar{\boldsymbol{\lambda}}' | (a_{n+1} a_1)(t) | I \rangle}{\langle \bar{\boldsymbol{\lambda}}' | I \rangle} + \frac{i}{L} \left[ K(k_0') e^{8it \sin^2(k_0'/2)} e^{ik_0' n} - K(k_0) e^{8it \sin^2(k_0/2)} e^{ik_0 n} \right] . \tag{E.13}$$

This shows that the two choices of representative state lead to *different* results in the thermo-dynamic limit, which generally does not even exist as $K(k_0) \propto L^m$. This shows that for this particular initial state a naive application of typicality ideas fails.

However, a few comments are in order. First, since $\rho_h(k) \sim k^{2m}$ at small $k$, the smallest hole in a representative state $\boldsymbol{\lambda}$ is typically of order $L^{-1/(2m+1)}$, and in this case the additional terms are in fact negligible. Hence for such "typical" states, typicality ideas can be applied. This fact is confirmed numerically by observing that when one averages (E.9) over representative states, one indeed recovers (E.5). Second, in the problem at hand one can slightly change the initial state by imposing for example $K(k) = K(\delta)$ for $k < \delta$ for a fixed small $\delta$. With this "regularisation" one obtains (E.11), which is now well-behaved and allows for the limit $\delta \to 0$ to be taken. In this limit one recovers the expected result (E.5).

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
