# Peer review of "Systematic strong coupling expansion for out-of-equilibrium dynamics in the Lieb-Liniger model"

_SciPost Physics, doi:SciPost Phys. 11, 068 (2021)_

## Round 1 · Referee Report · Anonymous (Referee 1) · 2021-4-12

Report

The authors address the exact computation of observables after a quantum quench in interacting integrable systems. They focus on a systematic strong-coupling expansion, which was already introduced by the authors in previous works. Here, for the first time, it is applied to the time evolution after a quench. The authors study the prototypical example of the Lieb-Liniger model, although the logic is more general and works, in principle, for more other systems and quenches.

Although necessarily technical, the paper is very well written. I appreciate that the main results are nicely presented in a dedicated section, and that the main formulas are illustrated with plots. Subsequent technical parts are also clearly organized and the discussion is always easy to follow.

The material and the research presented is timely, given the interest in the nonequilibrium dynamics of isolated systems. Furthermore, I find the results very strong. As the authors stress, there exist very few cases where analytic results of any form can be obtained beyond the non-interacting limit. In fact, I am genuinely impressed by the calculations, and by how the authors manage to arrive at compact and nice results, despite the length of intermediate steps.

Overall, I don't have particular comments on the draft, as I think the presentation is already very good, and the authors already did a good job in presenting the relevant literature. Therefore, I essentially recommend publication of this excellent work as is. I only have a couple of minor questions/remarks for the authors, which I present in the following.

First, I find interesting the discussion around Fig. 3, regarding the absence of light-cones. I understand the justification given by the authors, I think it is sound. However, light-cone effects are usually more visible in the short-time regime, where, as the authors also mention, one might expect that a resummation of the perturbative series might be necessary in order to capture the relevant physics at finite interaction value. Therefore, is it possible that the absence of light-cones observed, e.g. in Fig. 3, is in fact an artifact of the truncation of the perturbative series? This would not be too surprising given that, contrary to the limit c=\infty, the energy of the initial state is finite for finite c.

I also have a few suggestions about references (however, I am not asking the authors to add them, if they think they are not relevant).

- In the conclusions, the authors mention that their work is the first one to obtain analytic results for the quench dynamics in a generic interacting integrable theory beyond the asymptotic late-time regime. I understand that here they refer to the dynamics of local observables, and I agree. More generally, however, analytic results have been obtained for other quantities after a quench, even in fully fledged interacting integrable systems such as the XXZ Heisenberg chain. One example is the real-time Loschmidt echo computed in

L. Piroli, B. Pozsgay, and E. Vernier, Nuclear Physics B 933, 454 (2018)

Furthermore, restricting to models with special limits or simple features, other examples exist. For instance, analytic results were also derived in

A. Cortés Cubero, J. Stat. Mech. 2016, 083107 (2016)

where exact results are obtained in the large-N limit of a matrix-valued quantum field theory. Finally, together with [85, 86] the authors might also consider adding a reference to

S. Sotiriadis, arXiv:2007.12683
S. Sotiriadis, arXiv:2010.03553

where the author also aims at a microscopic derivation of a GHD-like description of inhomogeneous quenches

  • validity: -
  • significance: -
  • originality: -
  • clarity: -
  • formatting: -
  • grammar: -

Author:  Etienne Granet  on 2021-06-25  [id 1528]

(in reply to Report 1 on 2021-04-12)

We thank the referee for their very positive comments about the draft.

Regarding the absence of lightcone: we agree that the series truncation might not be uniformly convergent near t=0, which would require resummation to get the correct t\to 0 behaviour of physical quantities. However, a lightcone would be observed in Fig 3 if there was a maximal quasiparticle velocity vmax, and so the connected correlator would be zero (or exponentially small) for all x> vmax t. This would thus be visible even for not small t, which is why we think this is not an artifact of the truncation of the series expansion.

Regarding the references: we focused indeed on results on local observables out-of-equilibrium. However we are happy to include the proposed references in the next version.

---

## Round 2 · Referee Report · Anonymous (Referee 2) · 2021-8-26

Report

The authors determine relaxation of three quantities of interest in the integrable Lieb-Liniger model. They employ a 1/c expansion in order to deal with the significant challenge to deal with the large spectral sums needed analyse time dependent quanties. In this way they arrive at leading orders exact expressions for the time evolution of (i) the one-point function of the interaction potential, (ii) the density-density correlation function and (iii) the steady-state expectation value of the two-point function of the interaction potential, all after a BEC groundstate quench.

The results in this paper are derived with a very high level of technical competency, and despite its technical nature is very well written. The paper will of broad interest to the quantum dynamics community. I recommend publication without further revision.

---

## Round 2 · Author Response

We thank the editor and the referee for the decision on the paper. We answered to the referee's question below their report. In the new version, we added the recommended references at the end of the first paragraph of the conclusion, which is the only modification.

Best regards,
Etienne Granet

---

## Editorial Decision

published